# PCAF-mediated acetylation regulates RAD51 dynamic localization on chromatin during HR repair

Jiajia Hou [1,2], Munan Shi[1,2], Jialu Hong[1], Yuting Liu[1], Xinyi Song[1], Haipeng Rao[1], Ying Ma [1], Chunchun Huang[1], Zhigang Hu [1], Lingfeng He[1], Zhigang Guo [1✉] & Feiyan Pan [1✉]

## Abstract

PCAF (p300-associated factor), a major histone acetyltransferase, is involved in many metabolic and pathogenic diseases. Here, we reveal a novel function of PCAF in homologous recombination repair (HR). We demonstrate that RAD51, a core protein in HR repair, physically interacts with the acetyltransferase domain of PCAF and is acetylated at lysine 40. This acetylation promotes RAD51 binding to ubiquitin, leading to its degradation via the ubiquitin–proteasome pathway. Following etoposide treatment, PCAF-induced acetylation removes RAD51 from chromatin to facilitate the late-phase HR processes. Overexpression of PCAF promotes premature dissociation of RAD51 from DNA damage sites. Notably, PCAF is downregulated in many cancers compared to adjacent tissues, correlating with shortened patient survival. Our findings suggest that decreased PCAF expression enhances HR efficiency, contributing to drug resistance in tumor cells, and the impact of PCAF on HR is dependent on its acetyltransferase activity. Our results highlight a novel role of PCAF in HR and provide a possible mechanism for tumor development and drug resistance caused by low expression of PCAF.

**Keywords** RAD51; PCAF; Homologous Recombination; Protein Stability
**Subject Categories** DNA Replication, Recombination & Repair; Post-translational Modifications & Proteolysis

## Introduction

DNA double-strand breaks (DSBs) are one of the most cytotoxic damages, posing a direct threat to genomic stability (Halazonetis et al, 2008). It can be induced by exogenous factors (e.g., radiation therapy or chemotherapy) or endogenous factors (e.g., reactive oxygen species and replication stress) (Deardorff et al, 2012). Defective DSB repair is implicated in numerous diseases, including cancer, immunodeficiency, and premature aging, thus cellular repair of DSBs needs to be performed in a timely, correct manner (White and Vijg, 2016). There are two main repair pathways for DSBs in eukaryotes, non-homologous end joining (NHEJ) and homologous recombination repair (Schwertman et al, 2016; Scully et al, 2019).

HR, which occurs mainly in the S and G2 phases of the cell cycle, uses a homologous DNA template to repair DSBs, thus maintaining genomic integrity and ensuring high-fidelity inheritance of genetic information, in contrast to the more error-prone NHEJ (Liu et al, 2021; Takata et al, 1998). The primary reaction in HR is the strand invasion and exchange between two homologous DNA molecules, which is catalyzed by the conserved RAD51/RecA family of proteins (Palacios-Blanco et al, 2024; Symington and Gautier, 2011; Wang et al, 2024). In this process, many proteins are regulated by post-translational modifications (PTM). For instance, lactylation of meiotic recombination 11 (MRE11) by the acetyltransferase CREB-binding protein (CBP) at the K673 site promotes its DNA binding, facilitating DNA end resection and HR (Chen et al, 2024). In addition, the ring type E3 ligase RNF19A mediated ubiquitination of BRCA1-associated ring domain-1 (BARD1) induces its dissociation from breast cancer susceptibility gene 1 (BRCA1) and inhibits HR activity (Zhu et al, 2021). Replication protein A (RPA) SUMOylation promotes the recruitment of RAD51 to DNA damage sites, thereby initiating DNA repair (Dou et al, 2010). Although the HR pathway has been well studied, the specific regulatory mechanism of the proteins involved in the HR process is still unclear.

RAD51, the eukaryotic homolog of the recombinase of *E. coli* RecA, is a 37 kDa protein consisting of an N-terminal (1–84 AA) and a C-terminal (84–339 AA) domain (Demeyer et al, 2021). Post-translational modifications of RAD51 have been extensively studied. RAD51 phosphorylation at T309 by checkpoint kinase 1 (Chk1) is crucial for its foci formation following hydroxyurea exposure (Sorensen et al, 2005). Moreover, the ring type E3 ligase RFWD3 promotes poly-ubiquitination of RAD51 both in vivo and in vitro, thus facilitating RAD51 degradation (Inano et al, 2017). In addition, SUMOylation of RAD51 at the K57 and K70 sites is essential for its chromatin recruitment and HR function (Hariharasudhan et al, 2022). Our previous studies have revealed that RAD51 is modulated by acetylation modifications (Shi et al, 2023). However, the acetyltransferases that mediate the acetylation of RAD51 and the role that acetylation of RAD51 plays in HR is still unknown.

Acetylation is a reversible post-translational modification, which is involved in the regulation of various biological processes such as

[1]Jiangsu Key Laboratory for Molecular and Medical Biotechnology, College of Life Sciences, Nanjing Normal University, 1 Wen Yuan Road, 210023 Nanjing, China. [2]These authors contributed equally: Jiajia Hou, Munan Shi. ✉E-mail: guo@njnu.edu.cn; panfeiyan@njnu.edu.cn

gene transcription, metabolism, and DNA damage repair (Shvedunova and Akhtar, 2022). p300-associated factor (PCAF) is a member of the acetyltransferase GNAT family. PCAF can acetylate lactate dehydrogenase chain B (LDHB) and promote inflammatory responses in non-alcoholic fatty liver disease (NALFD) processes (Wang et al, 2021). It also mediates the histone methyltransferase EZH2 acetylation and regulates its stability to promote lung adenocarcinoma progression (Wan et al, 2015). Besides this, PCAF also acetylates core histones, promoting transcriptional activation (Ogryzko et al, 1996). In addition, PCAF also functions as a fork-associated protein that promotes fork degradation in BRCA1-deficient cells (Kim et al, 2020). Several studies suggested that low expression of PCAF is associated with tumor progression and poor prognosis (Ma et al, 2024), but the underlying mechanism is still unclear.

In this study, we found that during HR, chromatin-bound RAD51 undergoes acetylation, leading to its dissociation from chromatin. The acetyltransferase PCAF interacts with the ATPase domain of RAD51 via its HAT domain, mediating the acetylation of RAD51 and promoting its degradation by the ubiquitin–proteasome pathway. We further demonstrated that overexpression of PCAF in cells causes RAD51 degradation, thereby suppressing the proper progression of HR, and this effect is correlated with its acetyltransferase activity.

## Results

### TSA treatment downregulates RAD51 and increases cell sensitivity to chemotherapeutic drugs

RAD51 is a core protein involved in homologous recombination repair. Previous studies have demonstrated that RAD51 undergoes acetylation, which regulates its protein stability (Shi et al, 2023). To investigate the dynamic changes in RAD51 acetylation during HR repair, we first examined the protein levels of RAD51 at different recovery time points following etoposide (ETO) treatment. Our results indicated that the total RAD51 protein levels peaked at 1 h and subsequently declined (Fig. EV1A). In contrast, RAD51 acetylation peaked at 2 h, coinciding with the observed decrease in protein levels (Fig. 1A), suggesting a potential link between RAD51 acetylation and protein level. To further explore this relationship, we fractionated cellular components and found a significant increase in chromatin-bound RAD51 following ETO treatment (Fig. 1B). Immunoprecipitation (IP) revealed that RAD51 acetylation was more pronounced in the chromatin-bond fraction than in the soluble fraction (Fig. 1C). These findings suggest that RAD51 undergoes acetylation primarily when bound to chromatin, and this modification may facilitate the subsequent reduction in RAD51 protein levels during HR.

To further elucidate the role of RAD51 acetylation in HR, we pretreated cells with histone deacetylase (HDAC) inhibitor Trichostatin A (TSA) and sirtuin inhibitor nicotinamide (NAM), observing that only TSA treatment elevated RAD51 acetylation levels (Fig. 1D). Treatment with the protein synthesis inhibitor cycloheximide (CHX) revealed that TSA accelerated the degradation of RAD51(Fig. 1E), indicating that acetylation reducing RAD51 protein stability. Previous study revealed that TSA-induced inhibition of DNA damage repair (Zhang et al, 2007).

Herein, we demonstrated that TSA treatment reduced the HR repair efficiency by approximately 50% through DR-GFP assay (Fig. 1F). Consistent with this, double-immunofluorescence staining further confirmed HR impairment, evidenced by increased foci of the DSB marker γH2AX and decreased foci of the HR repair marker RAD51 (Figs. 1G and EV1B). Notably, γH2AX foci persisted for up to 8 h in TSA-treated cells, suggesting delayed DSB repair and a compensatory accumulation of RAD51 foci (Fig. 1G). Given the strong correlation between HR efficiency and chemotherapeutic sensitivity, we next evaluated the impact of TSA treatment on cell sensitivity to chemotherapeutic agents. Cell survival assays revealed that TSA-treated cells exhibited significantly increased sensitivity to etoposide (ETO) and adriamycin (ADR) (Fig. 1H). Collectively, these data suggest that TSA treatment induces RAD51 acetylation and enhances cell sensitivity to chemotherapeutic agents by promoting RAD51 protein degradation.

### PCAF interacts with RAD51 and promotes its acetylation

To identify the acetyltransferase mediating RAD51 acetylation, we transfected several Flag-tagged acetyltransferases into HEK293T cells and performed IP using anti-RAD51 antibody. Western blot analysis revealed the presence of Flag-PCAF in the RAD51 precipitates, indicating an interaction between PCAF and RAD51 (Fig. 2A). In addition, we examined p300 in the immunoprecipitated of RAD51 and no specific band could be observed, excluding the possibility of p300/CBP involving in RAD51 acetylation (Fig. EV2A). Next, pull-down assay using M2 beads confirmed that Flag-PCAF could efficiently precipitate endogenous RAD51 (Fig. EV2B). IP-Western blot analysis further validated both the exogenous and endogenous PCAF-RAD51 interactions (Figs. 2B and EV2C). Moreover, the interaction persisted even after DNase I treatment, excluding the possibility of DNA-mediated interactions (Fig. EV2D). Further supporting this, we purified His-tagged protein and performed in vitro pull-down assays. The results showed that PCAF directly interacted with RAD51 in vitro (Figs. 2C and EV2E). Based on these findings, we hypothesized that PCAF may mediate RAD51 acetylation. To test this, we performed an in vitro acetylation assay, which confirmed that PCAF catalyzes RAD51 acetylation (Fig. 2D). Given the critical role of RAD51 acetylation in response to DNA damage, we next examined whether PCAF-RAD51 interaction is enhanced following DNA damage. As anticipated, the specific binding between these proteins was increased in ETO-treated cells (Fig. 2E). We also observed that following ETO treatment, PCAF was recruited to DSB sites and colocalized with RAD51, emphasizing its critical role in DSB repair (Fig. EV2F,G). These results suggest that the acetyltransferase PCAF directly interacts with RAD51 and promotes its acetylation, with the interaction being enhanced in response to DNA damage.

To further clarify the domains of RAD51 and PCAF responsible for their interaction, we first analyzed RAD51 structure and constructed truncated fragments. RAD51 R1 (residues 1–114) contains the N-terminal domain, while R2 (residues 82–339) comprises the ATPase domain (Morozumi et al, 2009). Co-IP demonstrated that PCAF interacts with both full-length RAD51 and its ATPase domain, but not with the N-terminal fragment (Fig. 2F). To further define the interacting region of PCAF, we

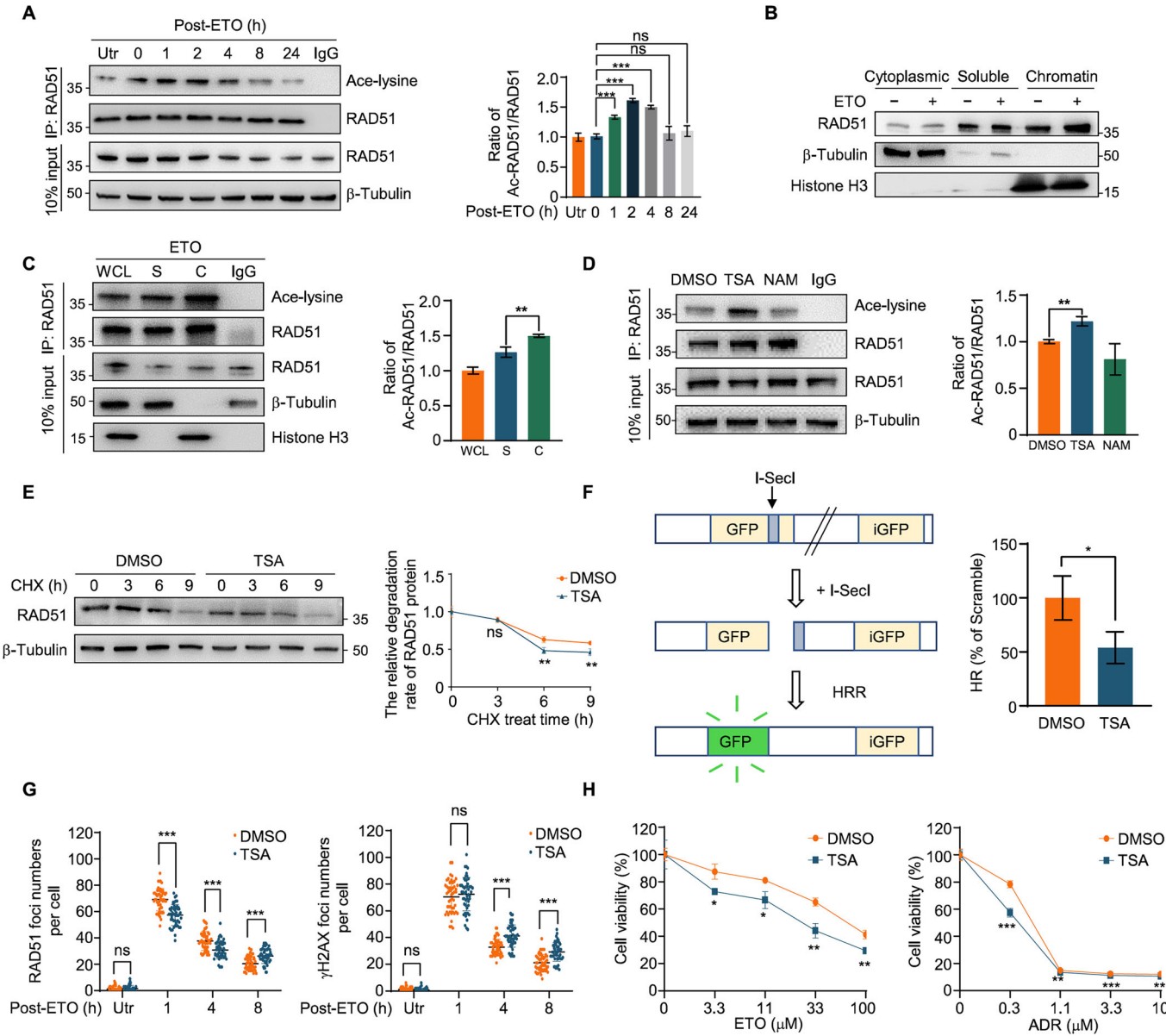

**Figure 1. TSA treatment downregulates RAD51 protein stability and inhibits HR.**

(A) Immunoblot of anti-RAD51 immunoprecipitates in HeLa cells untreated or treated with 20 μM ETO for 2 h and recovered at the indicated time points (left). Quantification of relative ac-RAD51 levels is shown (right). 0 h vs 1 h ($P = 0.0005$), 0 h vs 2 h ($P < 0.0001$), 0 h vs 4 h ($P < 0.0001$), 0 h vs 8 h ($P = 0.1967$), 0 h vs 24 h ($P = 0.5074$). (B) Immunoblot of RAD51 in HeLa cells, fractionated as indicated and treated with or without 20 μM ETO for 2 h. Fractions include cytoplasmic, non-chromatin-bound (soluble), and chromatin-bound (chromatin). (C) Immunoblot (left) and quantification (right) of ac-RAD51 in anti-RAD51 immunoprecipitates from HeLa cells, fractionated as indicated after treatment with 20 μM ETO for 2 h and recovered for 1 h. Fractions include whole cell lysates (WCL), non-chromatin-bound (soluble, S), and chromatin-bound (C). $P = 0.0064$. (D) Immunoblot (left) of ac-RAD51 in anti-RAD51 immunoprecipitates from HeLa cells treated with DMSO, TSA (0.5 μM, 6 h), and NAM (1 mM, 6 h). Quantification of relative ac-RAD51 levels is shown (right). $P = 0.0024$. (E) Immunoblot (left) and quantification (right) of RAD51 from HeLa cells pretreated with DMSO and TSA (0.5 μM, 6 h), followed by 100 μg/mL CHX exposure, with cell lysates collected at the indicated time points. 3 h ($P = 0.9540$), 6 h ($P = 0.009448$), 9 h ($P = 0.009947$). (F) Relative HR repair efficiency (right) in DMSO and TSA-treated (0.5 μM, 6 h) U2OS cells, as detected by HR reporter system (left). $P = 0.0338$. (G) Quantification of RAD51 (left) or γH2AX (right) foci from immunofluorescence in HeLa cells treated with or without 0.5 μM TSA for 6 h and recovered at the indicated points after ETO treatment (20 μM, 2 h) ($n = 50$). RAD51, Utr ($P = 0.447789$), 1 h ($P < 0.0001$), 4 h ($P < 0.0001$), 8 h ($P < 0.0001$). γH2AX, Utr ($P = 0.389794$), 1 h ($P = 0.429150$), 4 h ($P < 0.0001$), 8 h ($P < 0.0001$). (H) Cell survival assay in TSA-treated (0.5 μM, 6 h) HeLa cells in response to different doses of ETO (left) and ADR (right). ETO, 3.3 μM ($P = 0.013734$), 11 μM ($P = 0.015828$), 33 μM ($P = 0.004219$), 100 μM ($P = 0.003896$). ADR, 0.3 μM ($P = 0.000937$), 1.1 μM ($P = 0.002633$), 3.3 μM ($P = 0.0006$), 10 μM ($P = 0.009846$). All data are represented as mean ± SD of three independent experiments. $P$ values are from Student's $t$ tests (A, C–F, H) or Mann–Whitney $U$ test (G). *$P < 0.05$, **$P < 0.01$, ***$P < 0.001$, ns not significant. Source data are available online for this figure.

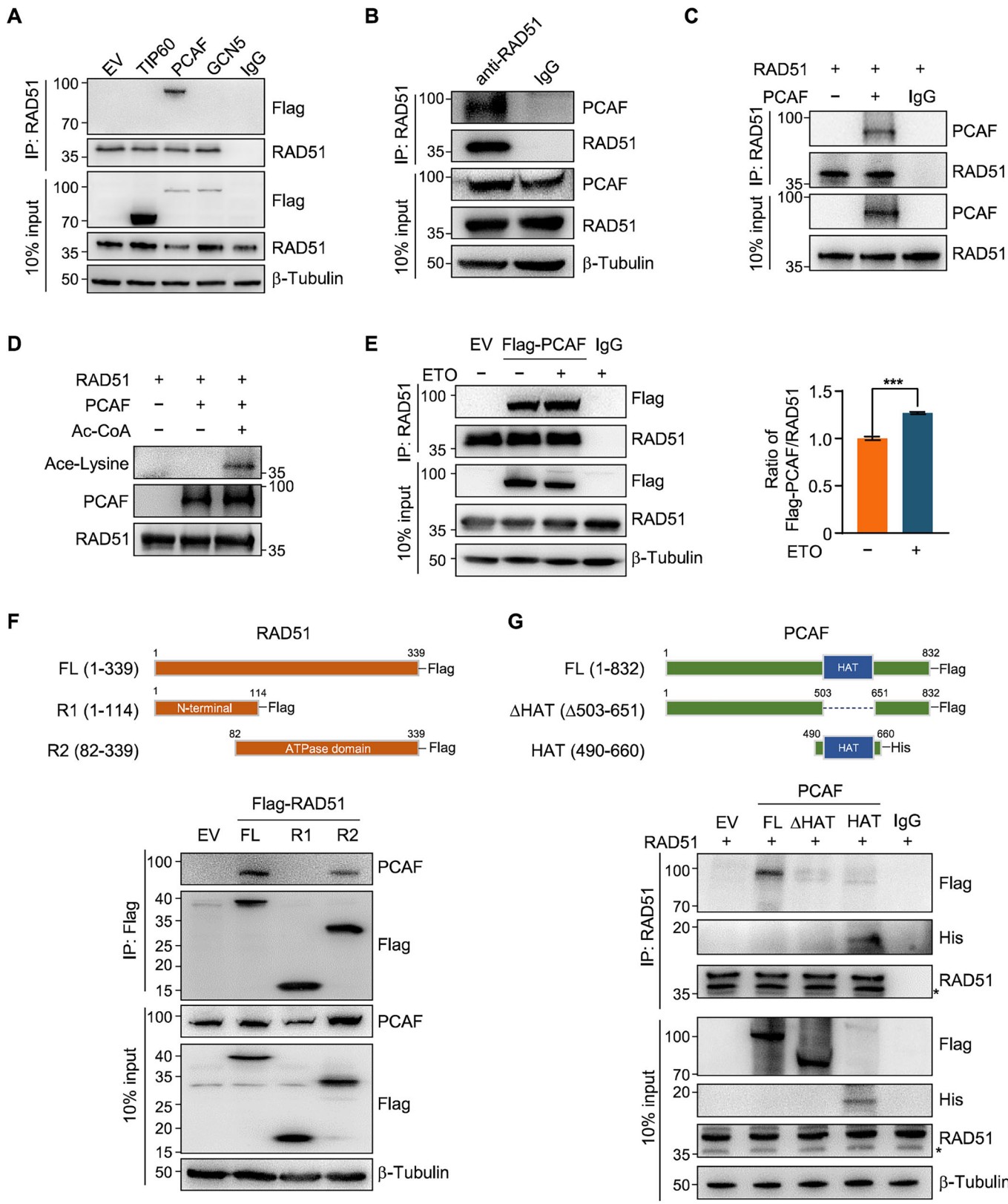

◄

**Figure 2. PCAF interacts with the ATPase domain of RAD51 via its HAT domain.**

(A) Immunoprecipitation to detect the interaction between RAD51 and different Flag-tagged HATs in HEK293T cells. (B) Immunoblot of PCAF in anti-RAD51 immunoprecipitates from HEK293T cells. (C) Interaction between purified His-RAD51 and His-PCAF. (D) Immunoblot of in vitro acetylation assay, purified RAD51 and PCAF were incubated in acetyltransferase assay buffer with or without Ac-CoA. (E) Immunoblot (left) and quantification (right) of Flag-PCAF in anti-RAD51 immunoprecipitates from HeLa cells transfected with empty vector or Flag-PCAF and treated with or without ETO (20 μM, 2 h). Data represented as mean ± SD of three independent experiments. P values are from Student's t tests. P < 0.0001. (F) Schematic of the RAD51 domains (upper). Immunoprecipitation was performed to determine the interaction and region between endogenous PCAF and the indicated Flag-RAD51 mutations (lower). (G) Schematic of the PCAF domains (upper). HAT Histone acetyltransferase. Immunoprecipitation was performed to determine the interaction and region between RAD51 and the indicated Flag-PCAF mutations. Asterisks indicate endogenous RAD51 (lower). Source data are available online for this figure.

constructed two truncations based on its domain structure: a short fragment containing the histone acetyltransferase domain (HAT) and a long fragment lacking this domain (ΔHAT) (Linares et al, 2007). Our results indicated that both the full-length PCAF and its HAT domain were detected in RAD51 immunoprecipitates, whereas deletion of the acetyltransferase domain completely abolished the interaction (Fig. 2G), suggesting that the HAT domain is crucial for PCAF's interaction with RAD51. Collectively, these findings confirm that PCAF interacts with the ATPase domain of RAD51 via its HAT domain.

## PCAF modulates RAD51 protein stability

Our previous study demonstrated that the acetylation of RAD51 modulates its protein stability. To assess whether the acetyltransferase PCAF participates in this regulation, we overexpressed PCAF in HEK293T and HeLa cells, which significantly decreased RAD51 protein levels (Figs. 3A and EV3A–C). This reduction was due to accelerated protein degradation, rather than transcription inhibition (Figs. 3B and EV3D). To further confirm PCAF's regulatory effect on RAD51 protein levels, we employed three sgRNA sequences to knock down PCAF, all of which increased RAD51 protein levels, with sgPCAF-3 exhibiting the most pronounced effect (Figs. 3C and EV3E). Given its high knockdown efficiency, sgPCAF-3 was selected for the following experiments. PCAF depletion enhanced the protein stability of RAD51 without affecting RAD51 transcription (Figs. 3D and EV3F), and PCAF re-expression reversed the RAD51 increase in knockdown cells (Fig. 3E). Taken together, these results suggest that PCAF regulates RAD51 protein stability.

There are two main pathways for protein degradation in eukaryotes, the ubiquitin–proteasome pathway and the lysosomal pathway (Gu et al, 2023). To identify the degradation pathway involved, we treated PCAF-overexpressing cells with the ubiquitin–proteasome inhibitor MG132 and the lysosomal inhibitors 3-Methyladenine (3-MA) and Bafilomycin A1 (Baf-A1). MG132 treatment fully restored RAD51 protein levels, whereas lysosomal inhibition had no effect (Figs. 3F and EV3A), indicating that PCAF promotes RAD51 degradation via the ubiquitin–proteasome pathway. Furthermore, PCAF overexpression increased both RAD51 acetylation and ubiquitination (Fig. 3G–I), while PCAF knockdown decreased these modifications (Fig. 3J–L). Collectively, these findings illustrate that PCAF induces RAD51 acetylation and promotes its degradation via the ubiquitin–proteasome pathway.

## K40 acetylation is essential for RAD51 function in HR

RAD51 acetylation occurs at lysine 40, 64, 70, 80, and 285 (Shi et al, 2023) and multiple sequence alignment revealed that these sites are highly conserved across species (Fig. EV4A). To further clarify which lysine site was significantly acetylated during HR, we generated arginine (R) mutants at each lysine to mimic the deacetylation state. We found that the acetylation levels of RAD51-K40R and RAD51-5KR (all five lysine sites mutated to arginine) were markedly decreased after ETO treatment (Fig. 4A), indicating that RAD51 is primarily acetylated at the K40 site during HR. However, the K70R point mutant was unstable and could not be expressed. It is noteworthy that a residual level of acetylation persists in RAD51-5KR, suggesting the existence of additional acetylation sites. Notably, compared to RAD51-WT, the acetylation level of K40R was significantly reduced in PCAF-overexpressed cells (Fig. 4B).

To further investigate the effect of RAD51 acetylation in HR, we constructed the lysine (K) to glutamine (Q) mutant K40Q to mimic constitutive acetylation. HR reporter assays in RAD51-knockdown U2OS cells showed that while RAD51-WT and K40R overexpression could rescue HR efficiency, K40Q could not (Fig. 4C). Notably, K40R exhibited higher HR efficiency than RAD51-WT. In addition, to further support the role of RAD51 acetylation in HR, we repeated the HR reporter assay and found that in cells expressing the RAD51 acetylation mutant, TSA treatment did not reduce HR efficiency as it did in wild type cells (Fig. EV4B). Next, we established RAD51 knockdown (RAD51 KD) HeLa cells using the CRISPR-Cas9 (Fig. EV4C) and transfected RAD51-WT, K40R, K40Q, and 5KR, respectively, into RAD51 KD cells. Following ETO treatment and 4-hour recovery, γH2AX and RAD51 protein levels and foci formation were assessed via western blot and immunofluorescence. RAD51 KD cells exhibited increased DNA damage, while expression of RAD51-WT, K40R, and 5KR restored RAD51 foci formation and reduced γH2AX foci numbers (Figs. 4D–F and EV4D–F). Among these mutants, RAD51-K40R and 5KR showed more RAD51 and less γH2AX foci numbers, suggesting enhanced DNA damage repair capacity (Figs. 4E and EV4E). We further examined cell survival under ETO and ADR treatment. The results showed that RAD51 KD significantly increased cell sensitivity to chemotherapeutic drugs, and RAD51-K40R and 5KR weaken this sensitivity (Figs. 4G and EV4G,H). Together, our results suggest that RAD51 acetylation at lysine 40 negatively regulates RAD51 function in HR, enhancing cellular sensitivity to DNA-damaging agents.

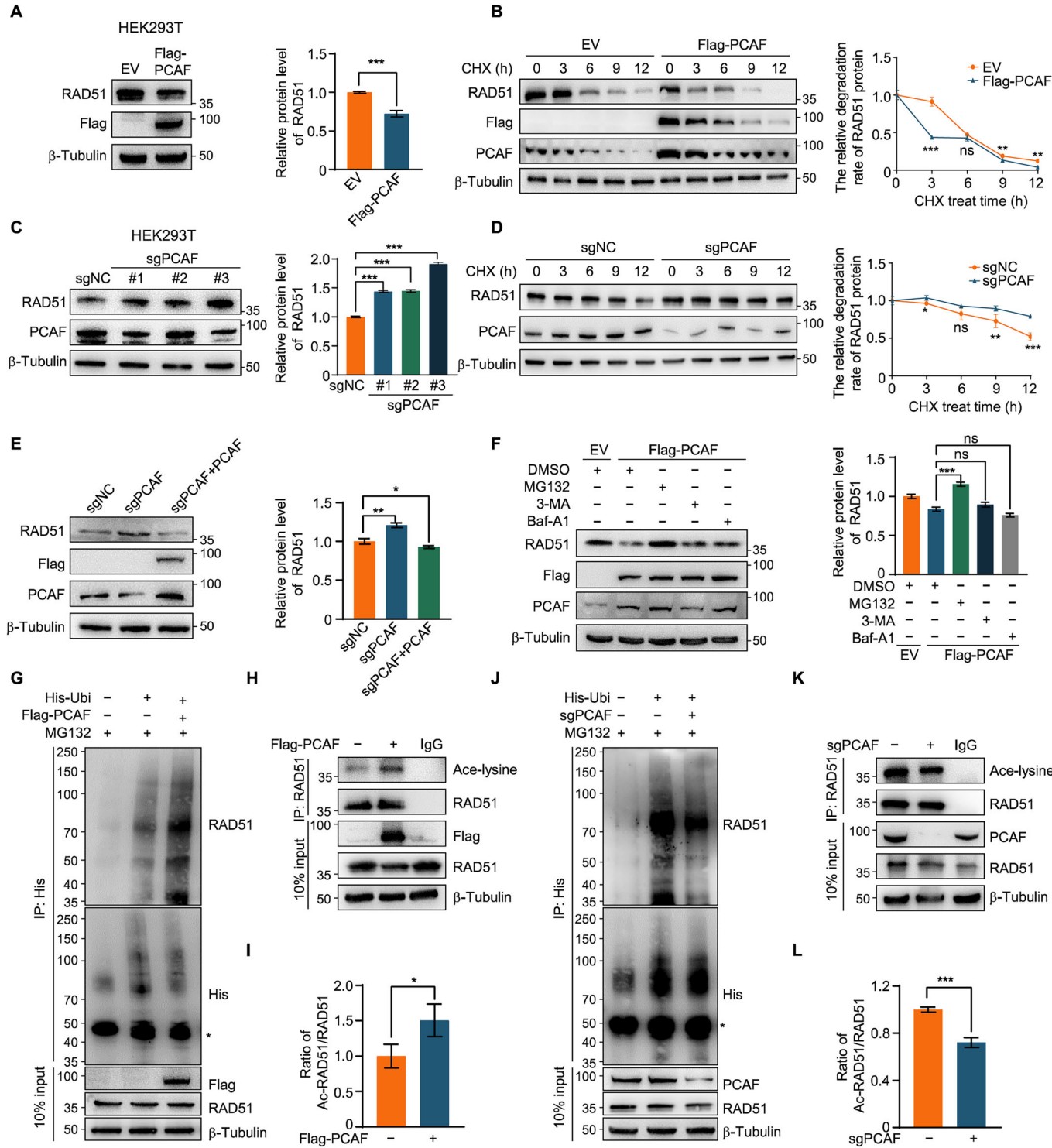

## PCAF inhibits HR by downregulating RAD51

To further characterize the physiological significance of PCAF in HR, we first analyzed the UALCAN (the University of Alabama at Birmingham CANcer data analysis portal) database and found that PCAF expression is low in multiple tumor types (Fig. 5A). Moreover, in certain cancers, including kidney renal clear cell carcinoma (KIRC)

and lung adenocarcinoma (LUAD), low PCAF expression is associated with poor patient survival (Fig. 5B), indicating that PCAF may be related to tumor resistance to chemotherapy. To test this hypothesis, we investigated the effect of PCAF on HR. Compared with control cells, PCAF overexpression resulted in decreased RAD51 protein levels, increased γH2AX levels (Fig. EV5A), and corresponding changes in RAD51 and γH2AX foci (Figs. 5C and EV5B). Notably,

**Figure 3. PCAF promotes RAD51 degradation via the ubiquitin–proteasome pathway.**

(A) Immunoblot (left) and quantification (right) of RAD51 in HEK293T cells overexpressing PCAF. $P = 0.0004$. (B) Immunoblot (left) and quantification (right) of RAD51 in HEK293T cells transfected with or without Flag-PCAF, treated with 100 μg/mL CHX and collected at the indicated time points. 3 h ($P = 0.000241$), 6 h ($P = 0.165788$), 9 h ($P = 0.003867$),12 h ($P = 0.001129$). (C) Immunoblot (left) and quantification (right) of RAD51 and PCAF in HEK293T cells transfected with sgNC or three different sgPCAF sequences for PCAF knockdown. sgNC vs sgPCAF#1 ($P < 0.0001$), sgNC vs sgPCAF#2 ($P < 0.0001$), sgNC vs sgPCAF#3 ($P < 0.0001$). (D) Immunoblot (left) and quantification (right) of RAD51 in HEK293T cells with PCAF knockdown using sgPCAF, treated with 100 μg/mL CHX and collected at indicated time points. 3 h ($P = 0.041424$), 6 h ($P = 0.111715$), 9 h ($P = 0.0039799$), 12 h ($P = 0.000993$). (E) Immunoblot (left) and quantification (right) of RAD51 in PCAF knockdown or PCAF rescued HEK293T cells. sgNC vs sgPCAF ($P = 0.0016$), sgNC vs sgPCAF+PCAF ($P = 0.0368$). (F) Immunoblot (left) and quantification (right) of RAD51 in PCAF-overexpressing HEK293T cells treated with different chemical reagents. DMSO vs MG132 ($P = 0.0001$), DMSO vs 3-MA ($P = 0.0762$), DMSO vs Baf-A1 ($P = 0.1$). (G) Immunoblot of Ubi-RAD51 in anti-His-Ubi immunoprecipitates from HEK293T cells transfected with His-Ubi and Flag-PCAF after treatment with 10 μM MG132 for 6 h. Asterisks indicate heavy chain. (H, I) Immunoblot (H) and quantification (I) of ac-RAD51 in anti-RAD51 immunoprecipitates from HEK293T cells transfected with empty vector or Flag-PCAF. $P = 0.0366$. (J) Immunoblot of Ubi-RAD51 in anti-His-Ubi immunoprecipitates from HEK293T cells transfected with His-Ubi and sgPCAF after treatment with 10 μM MG132 for 6 h. Asterisks indicate heavy chain. (K, L) Immunoblot (K) and quantification (L) of ac-RAD51 in anti-RAD51 immunoprecipitates from HEK293T cells transfected with sgNC or sgPCAF. Asterisks indicate heavy chain. $P = 0.0005$. All data are represented as mean ± SD of three independent experiments. $P$ values are from Student's $t$ tests. **$P < 0.01$, ***$P < 0.001$; ns not significant (A–L). Source data are available online for this figure.

γH2AX levels remained elevated at 8 and 24 h post-treatment, indicating impaired HR and persistent DNA damage, which resulted in compensatory increase in RAD51 foci. In contrast, PCAF knockdown obtained the converse result, further confirming the inhibitory effect of PCAF in HR (Figs. 5D and EV5C,D). Next, we performed the neutral comet assay to measure ETO-induced DSBs following overexpression or knockdown of PCAF. Tail moments analysis revealed that PCAF overexpression remarkably increased DSB accumulation, while PCAF knockdown had the reverse effect (Fig. 5E,F). This was further confirmed using an HR reporter assay (Fig. 5G). Importantly, ectopic expression of RAD51-WT in PCAF-overexpressed cells significantly enhanced HR (Figs. 5H,I and EV5E,F), suggesting that PCAF suppresses HR primarily through RAD51 downregulation. Cell survival assays further demonstrated that PCAF overexpression increased cell sensitivity to ETO and ADR, while ectopic expression of RAD51-WT eliminated this effect (Figs. 5J–L and EV5G–I). Altogether, these results suggest that PCAF suppresses HR by downregulating RAD51.

### PCAF function in HR depends on its HAT activity

To verify the importance of acetyltransferase activity for PCAF function in HR, we used PCAF-ΔHAT, a truncation lacking the acetyltransferase domain, and Anacardic Acid (PCAFi), a histone acetyltransferase inhibitor of PCAF for further investigation. After transfecting PCAF-WT and PCAF-ΔHAT into HEK293T cells, we found that PCAF-ΔHAT did not decrease RAD51 protein levels as PCAF-WT did (Fig. 6A), or affect RAD51 protein stability (Fig. 6B). Additionally, PCAF-ΔHAT did not affect the acetylation level and ubiquitin binding of RAD51 (Fig. 6C; Appendix Fig. S1A). Importantly, PCAFi treatment increased RAD51 protein levels in a dose-dependent manner and inhibited its acetylation (Appendix Fig. S1B,C). Taken together, these results suggest that the regulation of RAD51 by PCAF is correlated with its enzymatic activity.

Furthermore, we found that PCAF-WT could decrease the chromatin-bound RAD51 level following ETO treatment, while PCAF-ΔHAT could not (Fig. 6D). Electrophoretic mobility shift assay (EMSA) results suggested that RAD51 acetylation weakens its binding to overhang DNA, further supporting the role of PCAF-mediated acetylation in HR suppression (Appendix Fig. S1D). Additionally, PCAF-ΔHAT did not alter the formation of RAD51 and γH2AX foci (Fig. 6E; Appendix Fig. S1E), suggesting that deletion of the enzymatic domain significantly impacts PCAF's

function in DNA damage repair. This was further validated by the neutral comet assay (Fig. 6F). In addition, PCAFi treatment facilitated RAD51 foci formation and HR (Appendix Fig. S1F). In contrast, while PCAF overexpression significantly reduced HR efficiency, PCAF-ΔHAT had no effect (Fig. 6G; Appendix Fig. S1G). Cell survival assays also revealed that overexpression of PCAF increased cell sensitivity to ETO and ADR, whereas PCAFi treatment decreased drug sensitivity. Notably, PCAF-ΔHAT did not affect chemosensitivity (Fig. 6H,I; Appendix Fig. S1H). Collectively, our results confirm that PCAF inhibits DNA damage repair and enhances cellular sensitivity to chemotherapeutic agents through its enzymatic activity.

## Discussion

RAD51 plays a pivotal role in HR, not only because it promotes DNA homologous pairing and strand exchange, but also helps to reverse and remodel stalled replication forks (Bhowmick et al, 2022; Hanthi et al, 2024; Laurini et al, 2020; Ye et al, 2024). The function of RAD51 is tightly regulated by various post-translational modifications, including phosphorylation, ubiquitination, and SUMOylation (Demeyer et al, 2021). Herein, we demonstrate the role of RAD51 acetylation in HR and the detailed mechanism of which acetyltransferase regulates RAD51 acetylation.

Acetylation is an important post-translational modification of proteins, which was first discovered in histones that regulate gene transcription and later verified in non-histone proteins that regulate different cellular events. Histone acetylation is essential for genome stability, replication, and DNA damage repair (Dang and Wei, 2022; Shen et al, 2015). Multiple lines of evidence have shown that protein acetylation is highly related to DNA damage repair. Several HR proteins are known to be regulated by acetylation. The acetyltransferase TIP60-mediated acetylation of ATM is necessary for its kinase activity (Blackford and Jackson, 2017). Moreover, sirtuin 2 (SIRT2) facilitates HR by deacetylating BRCA1/BARD1 and enhancing the stability of the heterodimer, nuclear reservation, and localization to DNA damage sites (Minten et al, 2021). Our previous study showed that the deacetylase HDAC1 mediates RAD51 deacetylation and promotes HR (Shi et al, 2023). However, the mechanism underlying RAD51 acetylation during HR and the acetyltransferase responsible for its acetylation remains unclear. In this study, by measuring RAD51

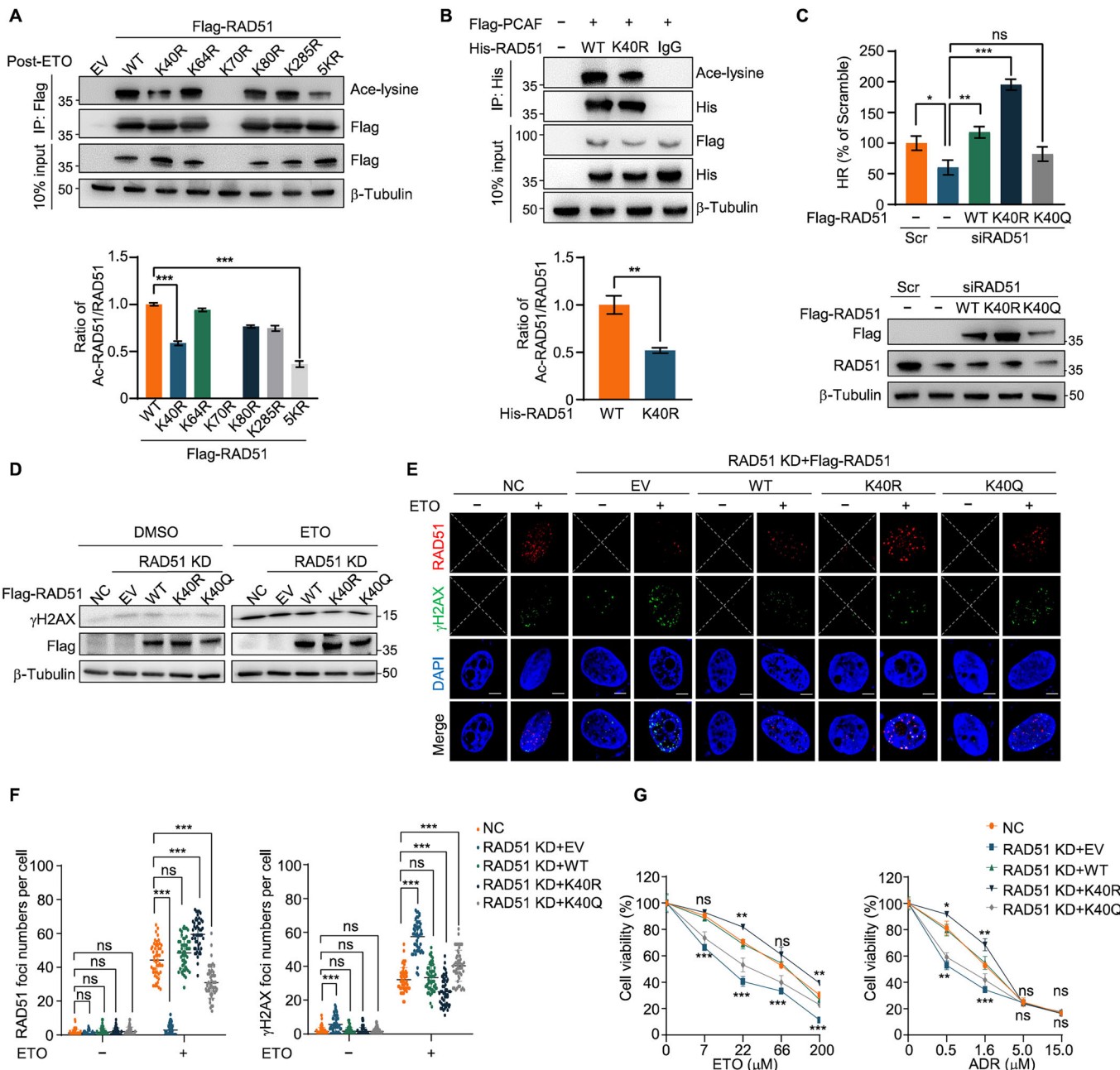

acetylation at different time points of HR, we found that the acetylation level of RAD51 was dynamically changed during HR and that chromatin-bound RAD51 was inclined to be acetylated (Figs. 1A–C and EV1A). Moreover, the peak time of acetylation coincides with the time point when total RAD51 and chromatin-bound RAD51 begin to decrease. This not only confirms the relationship between acetylation and protein reduction, but also suggests the important regulatory role of acetylation on RAD51 removal from chromatin. In fact, the localization of DNA repair proteins on chromatin is a dynamic process. They are precisely regulated to bind to chromatin at the right time and leave promptly after performing their function (Meir et al, 2015; Pinedo-Carpio et al, 2023; Zainu et al, 2024). It is reported that growth factor

receptor-bound protein 2 (GRB2) can bind and inhibit RAD51 ATPase activity to stabilize RAD51 on stalled replication forks (Ye et al, 2024). Rtt105, a yeast Ty1 transposon regulator, acts to stimulate Rad51 assembly and orchestrate RPA and Rad51 actions during HR (Wang et al, 2024). FIGNL1 interacting regulator of recombination and mitosis (FIRRM) directly binds to single-stranded DNA (ssDNA) and promotes RAD51 resolution during interstrand DNA cross-link (ICL) repair (Pinedo-Carpio et al, 2023). Notably, RFWD3 mediated ubiquitination contributes to the timely removal of RAD51 from DNA damage sites to facilitate HR (Inano et al, 2017). Consistent with their results, our data showed that acetylated RAD51 is more susceptible to ubiquitination and degradation, which means that our findings may provide a possible

◀ 

**Figure 4. Lys40 is a key acetylation site of RAD51.**

(A) Immunoblot (upper) and quantification (lower) of ac-RAD51 in anti-RAD51 immunoprecipitates from HeLa cells transfected with indicated Flag-RAD51 mutations after being treated with ETO (20 μM, 2 h) and recovered for 1 h. WT vs K40R ($P < 0.0001$), WT vs 5KR ($P < 0.0001$). (B) Immunoblot (upper) and quantification (lower) of ac-RAD51 immunoprecipitates from 293 T cells transfected with Flag-PCAF and His-RAD51 WT or K40R. $P = 0.0012$. (C) HR efficiency (upper) and immunoblot (lower) in U2OS cells transfected with scramble RNA or siRAD51 for 24 h, followed by transfection with the indicated Flag-RAD51 mutations for another 24 h. Scr vs siRAD51 ($P = 0.0149$), siRAD51 vs WT ($P = 0.0028$), siRAD51 vs K40R ($P < 0.0001$), siRAD51 vs K40Q ($P = 0.0877$). (D) Immunoblot of γH2AX in RAD51 KD HeLa cells transfected with indicated Flag-RAD51 mutations and treated with or without ETO (20 μM, 2 h) and recovered for 4 h. (E) Representative immunofluorescence images of RAD51 (red) and γH2AX (green) foci in RAD51 KD HeLa cells transfected with different Flag-RAD51 mutations and treated with or without ETO (20 μM, 2 h) and recovered for 4 h. Scale bars, 10 μm. X indicated that with the chosen microscopy settings, no signal was obtained. (F) Quantification of RAD51 (right) or γH2AX (left) foci per cell from (E) ($n = 50$). RAD51, ETO −, NC vs RAD51 KD + EV ($P = 0.0898$), NC vs RAD51 KD + WT ($P = 0.3234$), NC vs RAD51 KD + K40R ($P = 0.2311$), NC vs RAD51 KD + K40Q ($P = 0.1450$). RAD51, ETO +, NC vs RAD51 KD + EV ($P < 0.0001$), NC vs RAD51 KD + WT ($P = 0.0847$), NC vs RAD51 KD + K40R ($P < 0.0001$), NC vs RAD51 KD + K40Q ($P < 0.0001$). γH2AX, ETO −, NC vs RAD51 KD + EV ($P < 0.0001$), NC vs RAD51 KD + WT ($P = 0.94$), NC vs RAD51 KD + K40R ($P = 0.8332$), NC vs RAD51 KD + K40Q ($P = 0.7641$). γH2AX, ETO +, NC vs RAD51 KD + EV ($P < 0.0001$), NC vs RAD51 KD + WT ($P = 0.4152$), NC vs RAD51 KD + K40R ($P < 0.0001$), NC vs RAD51 KD + K40Q ($P < 0.0001$). (G) Cell survival assay was performed in RAD51 KD HeLa cells transfected with indicated Flag-RAD51 mutations in response to different doses of ETO (left) and ADR (right). ETO, NC vs RAD51 KD + K40R 7 μM ($P = 0.187613$), 22 μM ($P = 0.001712$), 66 μM ($P = 0.067985$), 200 μM ($P = 0.003486$). NC vs RAD51 KD + EV, 7 μM ($P < 0.0001$), 22 μM ($P = 0.000255$), 66 μM ($P < 0.0001$), 200 μM ($P = 0.000195$). ADR, NC vs RAD51 KD + K40R, 7 μM ($P = 0.030385$), 22 μM ($P = 0.004562$), 66 μM ($P = 0.555441$), 200 μM ($P = 0.645064$). NC vs RAD51 KD + EV, 7 μM ($P = 0.001492$), 22 μM ($P < 0.0001$), 66 μM ($P = 0.78915$), 200 μM ($P = 0.882987$). All data are represented as mean ± SD of three independent experiments. $P$ values are from Student's t tests (A–C, G) or Mann–Whitney U test (F). *$P < 0.05$, **$P < 0.01$, ***$P < 0.001$, ns not significant. Source data are available online for this figure.

explanation for the specific recognition of RAD51 by RFWD3. Moreover, whether knockdown of RFWD3 is additive or synergistic with PCAF and the detailed connection between acetylation and ubiquitination and the localization of these modifying enzymes on DNA at different time stages of HR require further investigation.

The major acetyltransferases include three families, p300/CREB-binding protein, MYST, and GNAT families (Zhao et al, 2018). We identified PCAF, a member of the GNAT family, as the acetyltransferase interacting with RAD51 (Fig. 2). Based on our previous results obtained from mass spectrometry analysis, we constructed point mutations to mimic the deacetylated status of RAD51 and found that PCAF-induced acetylation of RAD51 occurs predominantly at the lysine 40 (Fig. 4A,B). Compared with RAD51-WT, RAD51-K40R caused a significant decrease in γH2AX levels. Similar results were obtained through immunofluorescence assays when examining γH2AX foci (Fig. 4D,E). Furthermore, cell survival assays also showed that RAD51-K40R increased cellular resistance to the chemotherapeutic drugs ETO and ADR (Fig. 4F,G). Interestingly, we also found that the RAD51-5KR mutation still has some degree of acetylation, suggesting that there may be other sites involved in RAD51 acetylation, which requires further investigation (Fig. 4A). Although specific anti-K40 acetylation antibody is not yet available, the results of experiments with the mutant K40R strongly supported that lysine 40 is the candidate acetylated site. The development of specific antibodies is needed to confirm PCAF's role in RAD51 acetylation and to elucidate its physiological significance.

PCAF is involved in the regulation of many biological processes, including cell growth and differentiation, and transcriptional regulation (Lee et al, 2024). PCAF-mediated acetylation of pyruvate kinase M2 (PKM2) at lysine 305 leads to its autophagic degradation, regulating vascular endothelial cell aging (Wu et al, 2023). In addition, PCAF was associated with tumorigenesis and progression. In response to interleukin-1β (IL-1β) stimulation, PCAF translocated to mitochondria in cancer cells and protected tumor cells from iron metamorphosis (Han et al, 2023). In gastric cancer, low expression of PCAF correlates with poor clinicopathological features (Fei et al, 2016). Through UALCAN database analysis, we found that PCAF is low-expressed in many tumors and

correlated with poor prognosis (Fig. 5A,B). Our data demonstrate that PCAF is involved in HR by regulating RAD51 protein stability, thereby modulating cell sensitivity to chemotherapeutic reagents (Figs. 3 and 5). Since HR only occurs in the S and G2 phases, its efficiency is closely correlated with the cell cycle. In fact, the cell cycle is not responsible for the decreased HR efficiency induced by PCAF overexpression, because PCAF overexpression caused cell cycle arrest in the S and G2 phases and increased the ratio of these two phases (Appendix Fig. S1G) (Mateo et al, 2009). Meanwhile, based on the data obtained with the truncations lacking enzyme domain and PCAFi, we emphasized the critical role of acetyltransferase activity for PCAF in HR (Fig. 6). Moreover, our previous study found glyceraldehyde-3-phosphate dehydrogenase (GAPDH) as an essential HR regulator via HDAC1-dependent regulation of RAD51 stability. However, whether GAPDH works synergistically with PCAF still needs further investigation.

Kim et al demonstrated that PCAF participates in the HR process by acetylating histones, showing that PCAF knockout impairs HR and thereby identifying PCAF as a positive regulator of HR (Kim et al, 2019). In contrast, our study found that PCAF knockdown reduces RAD51 acetylation, leading to increased protein stability and enhanced HR, while PCAF overexpression enhances RAD51 acetylation and promotes its premature dissociation from chromatin, further highlighting the dynamic regulatory role of PCAF in HR proteins. Although there are some inconsistencies between our findings and those of Kim et al, they are not contradictory. It is well established that the dissociation of HR proteins from chromatin is essential for the completion of HR (Inano et al, 2017). In the absence of PCAF, RAD51 may fail to undergo acetylation and consequently be unable to dissociate from chromatin, thereby impeding HR progression. From this perspective, the findings of Kim et al indeed support the critical role of PCAF in HR. Importantly, our experiments involved PCAF knockdown rather than complete knockout, resulting in residual PCAF expression within the cells. This residual PCAF may be sufficient to facilitate RAD51 acetylation and its timely dissociation from chromatin. Therefore, it is plausible that an optimal level of PCAF is crucial for the proper completion of HR, as both excessive and insufficient PCAF could alter the chromatin binding dynamics

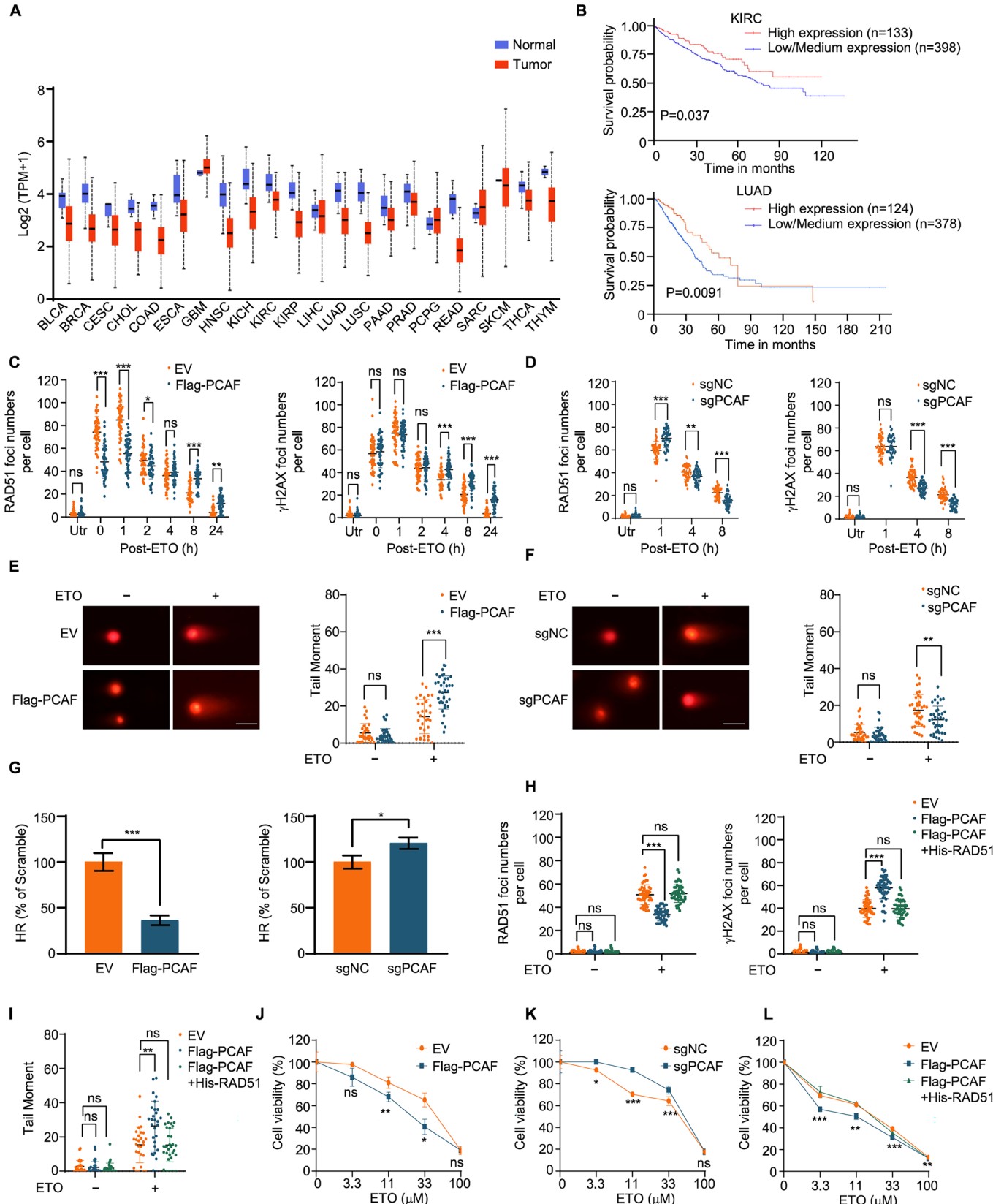

**Figure 5.  PCAF suppresses HR by downregulating RAD51.**

(A) Differences in PCAF expression levels between tumor and normal tissues across various cancers in the UALCAN database (https://ualcan.path.uab.edu/index.html). TPM Transcripts per million. (B) UALCAN database (https://ualcan.path.uab.edu/index.html) analysis of PCAF expression levels to patient survival of Kidney Renal Cell Carcinoma (KIRC, upper) and Lung Adenocarcinoma (LUAD, lower). (C) Quantification of RAD51 (left) or γH2AX (right) foci per cell from immunofluorescence in HeLa cells transfected with empty vector or Flag-PCAF and recovered at indicated points after ETO treatment (20 μM, 2 h) ($n = 50$). RAD51, Utr ($P = 0.81633$), 0 h ($P < 0.0001$), 1 h ($P < 0.0001$), 2 h ($P = 0.049434$), 4 h ($P = 0.129798$), 8 h ($P < 0.0001$), 24 h ($P < 0.0001$). γH2AX, Utr ($P = 0.610136$), 0 h ($P = 0.512304$), 1 h ($P = 0.616873$), 2 h ($P = 0.838133$), 4 h ($P < 0.0001$), 8 h ($P < 0.0001$), 24 h ($P < 0.0001$). (D) Quantification of RAD51 (left) or γH2AX (right) foci per cell from immunofluorescence in HeLa cells transfected with sgNC or sgPCAF and recovered at indicated points after ETO treatment (20 μM, 2 h) ($n = 50$). RAD51, Utr ($P = 0.306286$), 1 h ($P < 0.0001$), 4 h ($P = 0.001603$), 8 h ($P < 0.0001$). γH2AX, Utr ($P = 0.539115$), 1 h ($P = 0.946743$), 4 h ($P < 0.0001$), 8 h ($P < 0.0001$). (E) Neutral comet assay in HeLa cells transfected with indicated vectors and recovered for 4 h after ETO treatment (20 μM, 2 h). Scale bars, 100 μm. ETO– ($P = 0.142071$), ETO+ ($P < 0.0001$). (F) Neutral comet assay in HeLa cells transfected with sgNC or sgPCAF and recovered for 4 h after ETO treatment (20 μM, 2 h). Scale bars, 100 μm. ETO– ($P = 0.223976$), ETO+ ($P = 0.007230$). (G) Relative HR repair efficiency in U2OS cells transfected with Flag-PCAF (left) or sgPCAF (right). EV vs Flag-PCAF ($P = 0.0006$), sgNC vs sgPCAF ($P = 0.0201$). (H) Quantification of RAD51 (left) or γH2AX (right) foci per cell from immunofluorescence in HeLa cells overexpressing PCAF or co-transfected with Flag-PCAF and His-RAD51 with or without ETO treatment (20 μM, 2 h) and recovered for 4 h ($n = 50$). RAD51, ETO–, EV vs Flag-PCAF ($P = 0.7589$), EV vs Flag-PCAF+His-RAD51 ($P = 0.8566$). RAD51, ETO +, EV vs Flag-PCAF ($P < 0.0001$), EV vs Flag-PCAF+His-RAD51 ($P = 0.5383$). γH2AX, ETO–, EV vs Flag-PCAF ($P = 0.3643$), EV vs Flag-PCAF+His-RAD51 ($P = 0.8373$). γH2AX, ETO +, EV vs Flag-PCAF ($P < 0.0001$), EV vs Flag-PCAF+His-RAD51 ($P = 0.9586$). (I) The analysis of tail moment in HeLa cells overexpressing PCAF or co-transfected with Flag-PCAF and His-RAD51, with or without ETO treatment (20 μM, 2 h) and recovered for 4 h. ETO–, EV vs Flag-PCAF ($P = 0.4437$), EV vs Flag-PCAF+His-RAD51 ($P = 0.2682$). ETO +, EV vs Flag-PCAF ($P = 0.0021$), EV vs Flag-PCAF+His-RAD51 ($P = 0.9869$). (J) Cell survival assay was performed in HeLa cells transfected with empty vector or Flag-PCAF in response to different doses of ETO. 3.3 μM ($P = 0.078967$), 11 μM ($P = 0.003003$), 33 μM ($P = 0.010846$), 100 μM ($P = 0.958042$). (K) Cell survival assay was performed in HeLa cells transfected with sgNC or sgPCAF in response to different doses of ETO. 3.3 μM ($P = 0.016596$), 11 μM ($P = 0.000242$), 33 μM ($P = 0.031894$), 100 μM ($P = 0.929595$). (L) Cell survival assay in HeLa cells transfected with PCAF or co-transfected with Flag-PCAF and His-RAD51 in response to different doses of ETO. 3.3 μM ($P = 0.000127$), 11 μM ($P = 0.003017$), 33 μM ($P = 0.000569$), 100 μM ($P = 0.006091$). All data are represented as mean ± SD of three independent experiments. P values are from or Mann–Whitney U test (C–F, H, I) or Student's t tests (G, J–L). *$P < 0.05$, **$P < 0.01$ ***$P < 0.001$, ns not significant. Source data are available online for this figure.

of RAD51 and compromise HR efficiency. This precise regulation of protein levels is an important aspect that warrants further investigation.

In summary, our data demonstrate that acetylation regulates the dynamic balance of RAD51 protein levels. The acetyltransferase PCAF induces acetylation of RAD51 at lysine 40, facilitates its ubiquitin binding and subsequent degradation via the ubiquitin–proteasome pathway, while deacetylation stabilizes RAD51 in cells. When DSBs are generated, RAD51 binds to the 3'-overhanging ssDNA and interacts with PCAF, thus eliciting its premature dissociation from chromatin and decreased HR efficiency. Our study uncovers a novel role for PCAF in HR and provides a new mechanism for tumor development and chemotherapeutic drug resistance caused by aberrant PCAF expression.

# Methods

### Reagents and tools table

| Reagent/resource | Reference or source | Identifier or catalog number |
|---|---|---|
| **Experimental models** | | |
| HEK293T | ATCC | CRL-1573 |
| HeLa | ATCC | CCL-2 BSL 2 |
| U2OS-HR reporter system | Liu Songbai's group | N/A |
| **Recombinant DNA** | | |
| Flag-RAD51 | Vigene Biosciences | N/A |
| Flag-TIP60 | CORUES | N/A |
| Flag-GCN5 | CORUES | N/A |
| Flag-PCAF | CORUES | N/A |
| Flag-RAD51-K40R | This study | N/A |
| Flag-RAD51-K64R | This study | N/A |
| Flag-RAD51-K70R | This study | N/A |
| Flag-RAD51-K80R | This study | N/A |

| Reagent/resource | Reference or source | Identifier or catalog number |
|---|---|---|
| Flag-RAD51-K285R | This study | N/A |
| Flag-RAD51-5KR | This study | N/A |
| His-RAD51 | This study | N/A |
| His-PCAF | CENCEFE Biotech | N/A |
| RAD51-R1 | This study | N/A |
| RAD51-R2 | This study | N/A |
| PCAF-HAT | This study | N/A |
| PCAF-ΔHAT | This study | N/A |
| **Antibodies** | | |
| Anti-Acetylated-Lysine (Rabbit polyclonal) | Cell Signaling Technology | Cat#9441S |
| Anti-Phospho-Histone H2AX Ser139 (Mouse monoclonal) | Cell Signaling Technology | Cat#80312S; |
| Anti-RAD51 (Rabbit polyclonal) | Abcam | Cat#ab133534 RRID: AB_2722613 |
| Anti-Histone H3 (Rabbit polyclonal) | Abcam | Cat#ab18521 RRID: AB_732917 |
| Anti-p300/KAT3B (Mouse polyclonal) | Abcam | Cat#ab3864 |
| Anti-His tag (Mouse polyclonal) | Abclonal | Cat#AE003 RRID: AB_2728734 |
| Anti-PCAF (Rabbit polyclonal) | Abclonal | Cat#A0066 RRID: AB_2756928 |
| HRP Goat Anti-Rabbit IgG | Abclonal | Cat#AS014 RRID: AB_2769854 |
| HRP Goat Anti-Mouse IgG | Abclonal | Cat#AS003 RRID: AB_2769851 |
| Anti-RAD51 (Rabbit polyclonal) | Proteintech | Cat#14961-1-AP RRID: AB_2177083 |

| Reagent/resource | Reference or source | Identifier or catalog number |
|---|---|---|
| Anti-DYKDDDDK tag (Rabbit polyclonal) | Proteintech | Cat#20543-1-AP RRID: AB_11232216 |
| Anti-TBB5 (Mouse monoclonal) | Abgent | Cat#AM1031A RRID: AB_1968384 |
| Alexa Flour™ 488 donkey anti-mouse IgG (H + L) | Thermo Fisher Scientific | Cat#A21202 RRID: AB_141607 |
| Alexa Flour™ 594 donkey anti-rabbit IgG (H + L) | Thermo Fisher Scientific | Cat#A21207 RRID: AB_141637 |
| **Oligonucleotides and other sequence-based reagents** | | |
| sgRNA-RAD51 | TGTTGCCTATGCGCCAAAGA | |
| sgRNA-PCAF #1 | TATTTGCTGCAGGTCGGCTC | |
| sgRNA-PCAF #2 | TCAGTCTAACAGAATCCTGT | |
| sgRNA-PCAF #3 | CGGAGTTGTAGCCATGCCCT | |
| Scramble-siRNA-F | 5'-ACUCCCCUUGCUCAUGUAUACTT-3' | |
| Scramble-siRNA-R | 5'-GUACAUGAGCAAGGGGAGUTT-3' | |
| RAD51-siRNA-F | 5'-AGCUUCUUCCAAUUUCUUCTT-3' | |
| RAD51-siRNA-R | 5'-GAAGAAAUUGGAAGAAGCUTT-3' | |
| qRAD51-F | 5'-GTACATTGACACTGAGGGTACC-3' | |
| qRAD51-R | 5'-CTTGATAAAGGAGCTGGGTCTG-3' | |
| qActin-F | 5'-ACATCCGCAAAGACCTGTAC-3' | |
| qActin-R | 5'- TGATCTTCATTGTGCTGGGTG-3' | |
| RAD51-K40R-F | 5'-TGTGAAGAGATTGGAAGAAGCTGGATTCC-3' | |
| RAD51-K40R-R | 5'-CCAATCTCTTCACATCGTTGGCATTTATG-3' | |
| RAD51-K64R-F | 5'-TGCGCCAAAGAAGGAGCTAATAAATATTAGGGGAATTAGTGAAGC-3' | |
| RAD51-K64R-R | 5'-GCTTCACTAATTCCCCTAATATTTATTAGCTCCTTCTTTGGCGCA-3' | |
| RAD51-K70R-F | 5'-AAGCCAGAGCTGATAAAATTCTGGCTGAGGC-3' | |
| RAD51-K70R-R | 5'-CAGCTCTGGCTTCACTAATTCCCTTAATATT-3' | |
| RAD51-K80R-F | 5'-TCTGGCTGAGGCAGCTAGATTAGTTCCAATGGGTT-3' | |
| RAD51-K80R-R | 5'-AACCCATTGGAACTAATCTAGCTGCCTCAGCCAGA3' | |
| RAD51-K285R-F | 5'-CGATGTTTGCTGCTGATCCCAAAAGACCTATTGGAGGAA-3' | |
| RAD51-K285R-R | 5'-TTCCTCCAATAGGTCTTTTGGGATCAGCAGCAAACATCG-3' | |
| RAD51-R1-F | 5'-TACCGAGCTCGGATCCCATGGCAATGCAGATGCAG-3' | |
| RAD51-R1-R | 5'-TGGATCCTTATCGTCGTCATCCTTGTAATCTTGAAGTAGTTTGTC-3' | |
| RAD51-R2-F | 5'-TACCGAGCTCGGATCCCATGGTTCCAATGGGTTTCACC-3' | |
| RAD51-R2-R | 5'-GTGGATCCTTATCGTCGTCATCCTTGTAATCGTCTTTGGCATCTCC-3' | |
| PCAF-ΔHAT-F | 5'-TCACGTGGTTGGCAATTCCCTCAACCAGAAACCAAACAAGAAGA-3' | |
| PCAF-ΔHAT-R | 5'-TCCATGGGGAACCTTATAACTTCATAATATCCTGGAGCTTCTGTTC-3' | |
| PCAF-HAT-F | 5'-TACCGAGCTCGGATCCCATGCGCAGGGGTGTAATTGAATTTCACG-3' | |
| PCAF-HAT-R | 5'-TGGATCGTGGTGATGGTGATGATGAGAAATTCTGTGTACGGGAT-3' | |
| **Chemicals, enzymes and other reagents** | | |
| DMEM | Keygen Biotech | KGL1206-500 |
| Fetal Bovine Serum | Biochannel | BC-SE-FBS07 |

| Reagent/resource | Reference or source | Identifier or catalog number |
|---|---|---|
| Etoposide | Selleck | S1225 |
| MG132 | MedChem Express | HY-13259 |
| Trichostatin A | MedChem Express | HY-15144 |
| Nicotinamide | MedChem Express | HY-B0150 |
| Anacardic Acid | MedChem Express | HY-N2020 |
| Doxorubicin | MedChem Express | HY-15142A |
| 2 × Phanta Flash Master Mix | Vazyme | P510-01 |
| IP lysis buffer | beyotime | P0013 |
| cocktail | MedChem Express | HY-K0010 |
| protein A/G magnetic beads | Selleck | B23201 |
| M2 beads | Sigma | M8823 |
| cell counting-kit8 | APE-Bio | K1018 |
| FreeZol Reagent | Vazyme | R711-01 |
| HiScriptQ RT SuperMix | Vazyme | R123-01 |
| ChamQ SYBR qPCR Master Mix | Vazyme | Q341-02 |
| ExFect Transfection Reagent | Vazyme | T101-01 |
| His-tag purification kit | Beyotime | P2226 |
| BamHI | Takara | 1010A |
| HindIII | Takara | 1060A |
| **Software** | | |
| GraphPad Prism 8.0 | GraphPad Software | |
| ImageJ | NIH | |

## Cell culture

The human embryonic kidney cell line HEK293T and human cervical cancer cell line HeLa cells were purchased from ATCC, U2OS cells containing HRR reporter system and pCAGGS-I-SceI plasmid were gifts from Liu Songbai's group at Suzhou Vocational Health College. RAD51 knockdown U2OS cells were obtained using transfection with siRNA and the efficiency was detected by western blotting. RAD51 knockdown HeLa cells were generated by CRISPR/Cas9, using pCas-BSD-U6 containing a gDNA targeting RAD51 (5′-TGTTGCCTATGCGCC AAAGA-3′). Transient transfection of sgRNA was used to knock down PCAF in HeLa, HEK293T, and U2OS cells. These cells were cultured in Dulbecco's modified Eagle's (DMEM) supplemented with 10% fetal bovine serum.

## Chemicals

Etoposide was applied at a final concentration of 20 μM for 2 h. MG132 treatment was performed for 6 h with a final concentration of 10 μM. TSA treatment was performed for 6 h with a final concentration of 0.5 μM. NAM treatment was performed for 6 h with a final concentration of 1 mM. Anacardic Acid treatment was performed for 24 h with a final concentration of 40 μM.

## Plasmid construction and transfection

Flag-RAD51 cloned into p-ENTER plasmid was purchased from Vigene Biosciences. Flag-TIP60, Flag-GCN5, and Flag-PCAF cloned into the pcDNA3.1 plasmid were purchased from CORUES.

His-PCAF cloned into pET-28a was purchased from CENCEFE Biotech. Flag-RAD51 point mutant plasmid K40/64/70/80/285R/ 5KR and His-RAD51 cloned into pET-28a were generated using 2× Phanta Flash Master Mix, and the primers used are listed in Reagents and Tools Table. RAD51 and PCAF truncated fragments were cloned in plasmid pcDNA3.1, and the sequences were cloned into the BamHI and HindIII sites of the vector.

## Western blot

Cells were washed three times with PBS and lysed with RIPA on ice, and the samples were sonicated three times using an ultrasonic crusher. After this, the samples were boiled in a boiling water bath for 5 min. The proteins were separated using 8–12% SDS-PAGE and transferred onto the PVDF membrane. The membrane was blocked with 5% skimmed milk for 1.5 h. After blocking, the membrane was incubated with the corresponding primary antibody and kept at 4 °C overnight. After washing the membrane three times with PBST, the corresponding secondary antibody was incubated at room temperature for 1 h, after incubation, the membrane was washed three times with PBST. The chemiluminescent solution was prepared according to the instructions and then added to the PVDF membrane in a homogeneous drop. Images were scanned with a Tanon 4500 imaging system (China) and quantified with ImageJ (NIH).

## Immunoprecipitation

Cells were washed three times with ice-cold PBS and lysed with IP lysis buffer, and the protease inhibitors PMSF and cocktail were also added. Cell lysates were collected and sonicated 3 times using the ultrasonic breaker and then placed on the rotator at 4 °C for 3 to 4 h for lysis. Centrifuged the lysate at 4 °C, 12,000 rpm for 10 min and collected the supernatant. 10% of the supernatant was taken out as an input group, and the rest was divided into two tubes. One tube was added with protein A/G magnetic beads conjugated with the corresponding antibody or M2 beads, and the other tube was added with IgG as control, then placed on a rotator at 4 °C overnight to ensure enough binding between the protein and the magnetic beads. The magnetic beads were washed three times with PBS containing PMSF and cocktail to eliminate non-specifically bound proteins and contaminants. The proteins were detected using immunoblotting.

## Cell survival assay

Cell survival was detected by cell counting kit-8. Cells were seeded in 96-well plates at a density of 3000 cells per well, and after pretreatment of the cells according to the assay, ETO or ADR was added according to the gradient of concentration. After 48 h of treatment, 10 μL of CCK-8 solution was added to each well, and incubated in the incubator for 2 h. A microplate reader (TECAN Infinite F200 PRO, Switzerland) detected absorbance values at 450 nm.

## Reverse transcription and quantitative real-time PCR

Total RNA was extracted from cells using FreeZol Reagent. 0.5-1 μg of total RNA was used as the template for reverse transcription by HiScriptQ RT SuperMix for qPCR. Quantitative

PCR (qPCR) was performed using ChamQ SYBR qPCR Master Mix. Primers for human RAD51 and Actin are shown in Reagents and Tools Table.

## Immunofluorescence

Cells were seeded in 12 wells at a density of 30,000 cells per well. Cells were washed with PBS three times and fixed with 4% paraformaldehyde for 30 min. Using 0.5% TritonX-100 to break the cell membrane for 10 min, followed by two hours of blocking with 3% BSA. After blocking, primary antibodies were prepared using BSA and incubated overnight at 4 °C. Then cells were washed three times with PBST and incubated with fluorescent secondary antibody for two hours at room temperature. Cells were then stained with DAPI, observed and photographed by laser confocal microscope (Ti-E-A1R, Nikon, Japan). Within the same experiment, the exposure settings for identical fluorescence channels were kept consistent across groups to enable accurate comparisons of signal intensity. However, in certain groups, the expression level of the target protein was extremely low, resulting in undetectable signals in the images.

## HR efficiency system

HR efficiency was measured using DR-GFP cells. Cells were seeded into six-well plates, and when the density grew to 70%, Flag-PCAF, ΔHAT, sgPCAF, Flag-RAD51 WT, K40R, K40Q, or siRNAs were transfected into DR-GFP U2OS cells using ExFect Transfection Reagent. After 12 h of transfection, the I-SceI plasmid was transfected using ExFect Transfection Reagent. The cells were harvested after 48 h and analyzed for GFP expression by flow cytometry (FACSCalibur, BD Biosciences). Data were analyzed using CellQuest Pro (BD Biosciences).

## Neutral comet assay

In the neutral comet assay, cells were harvested at different times of recovery after the ETO treatment. First, a layer of normal melting point agarose gel was dropped onto the slide. After it solidified, the cells were mixed with 0.7% low melting point agarose gel and then applied on top of the solidified layer. Then transfer the slides to a cassette containing pre-cooled cell lysis working solution and leave at 4 °C for 1 h. After discarding the lysate, PBS was added and left for 3 min. Place the slide in a horizontal electrophoresis tank and slowly add cold neutral electrophoresis solution, leaving it for 1 h to deconvolute the DNA. After deconvolution, perform electrophoresis at 25 volts for 30 min. Following electrophoresis, place the slides into the cassettes, add the neutralization solution, and keep them at 4 °C to neutralize three times for 10 min each. Then add 10 μL of diluted PI to each well and incubate for 15 min at room temperature. Each slide was photographed by the Zeiss Axiovert 200 M microscope and comet tail analysis was calculated using the OpenComet plugin from ImageJ.

## Chromatin fraction extraction

First, cells were lysed with E1 buffer (50 mM HEPES-KOH, pH 7.5, 140 mM NaCl, 1 mM EDTA, pH 8.0, 10% glycerol, 0.5% NP40, 0.25% Triton X-100, 1 mM DTT) containing 1× protease inhibitor cocktail. After the centrifugation ($1100 \times g$ for 2 min at 4 °C), the

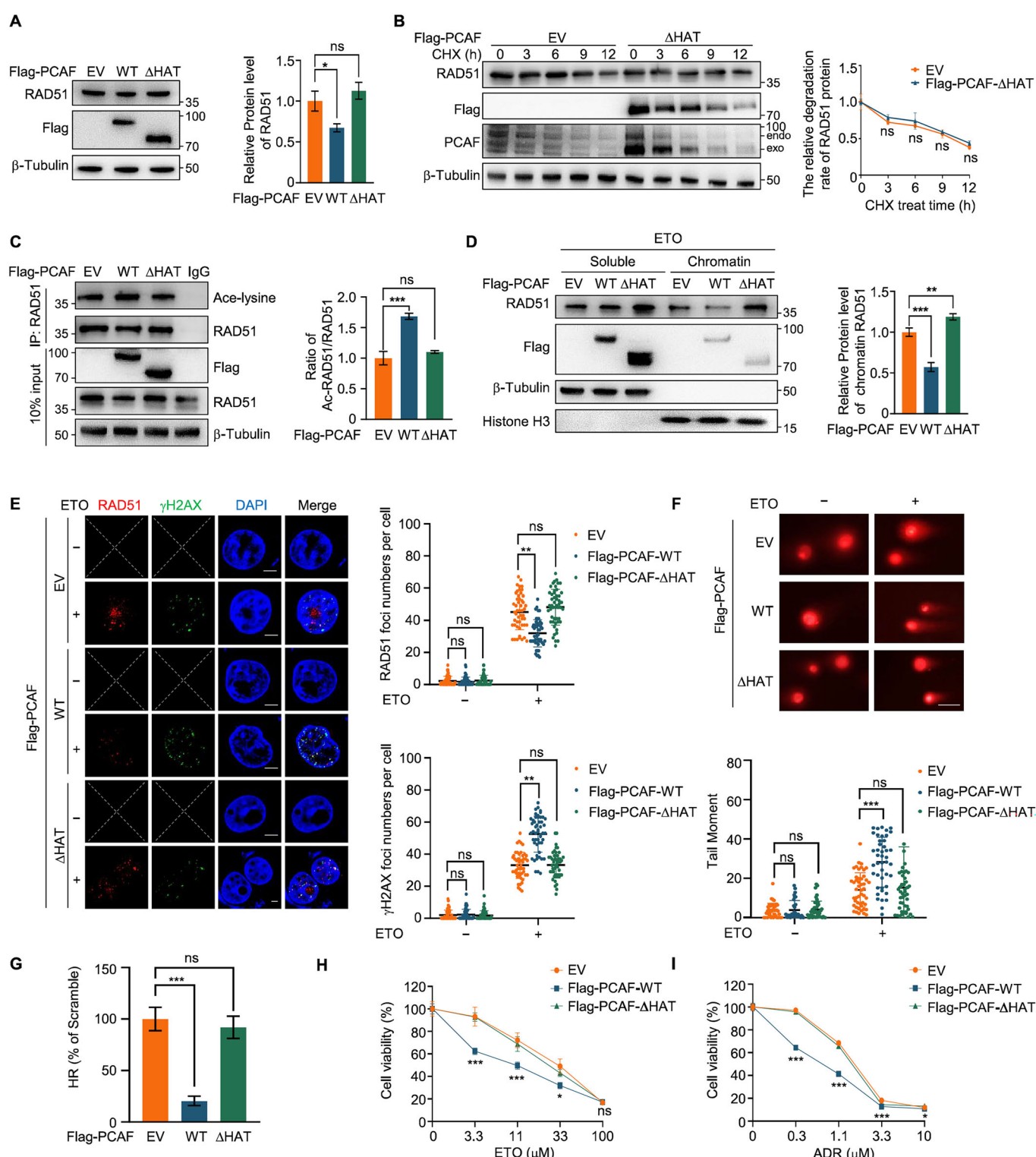

supernatant was collected as cytoplasmic. The pellet was washed two times with E1 buffer, then lysed with E2 buffer (10 mM Tris-HCl, pH 8.0, 200 mM NaCl, 1 mM EDTA, pH 8.0, 0.5 mM EGTA, pH 8.0) supplemented with 1× protease inhibitor cocktail. After the centrifugation (1100 × g for 2 min at 4 °C), the supernatant was

collected as nuclear soluble. The pellet was washed two times with E2 buffer, then lysed with E3 buffer (500 mM Tris-HCl, pH 6.8, 500 mM NaCl, 1× protease inhibitor cocktail) for Western blot or E3 (benzonase) buffer (50 mM Tris-HCl, pH 7.5, 20 mM NaCl, 1 mM MgCl₂, 1% NP40, 1× protease inhibitor cocktail) for pull

◀ **Figure 6. The HAT activity of PCAF is important for HR.**

(A) Immunoblot (left) and quantification (right) of RAD51 in HEK293T cells transfected with indicated Flag-PCAF variants. EV vs WT ($P = 0.0126$), EV vs ΔHAT ($P = 0.2355$). (B) Immunoblot (left) and quantification (right) of RAD51 in HeLa cells transfected with empty vector or Flag-PCAF-ΔHAT, treated with 100 μg/mL CHX, and collected cell lysates at indicated time points. 3 h ($P = 0.084659$), 6 h ($P = 0.386378$), 9 h ($P = 0.483797$), 12 h ($P = 0.051411$). (C) Immunoblot (left) and quantification (right) of ac-RAD51 in anti-RAD51 immunoprecipitates from HeLa cells transfected with indicated Flag-PCAF variants. EV vs WT ($P = 0.0006$), EV vs ΔHAT ($P = 0.1937$). (D) Immunoblot (left) and quantification (right) of RAD51 in non-chromatin-bound (soluble) and chromatin fractions of HeLa cells transfected with the indicated vectors and treated with ETO (20 μM, 2 h) and recovered for 1 h. EV vs WT ($P = 0.0006$), EV vs ΔHAT ($P = 0.0075$). (E) Representative immunofluorescence images (left) and quantifications (right) of RAD51 (red) and γH2AX (green) foci in HeLa cells transfected with indicated Flag-PCAF variants and treated with or without ETO (20 μM, 2 h). Scale bars, 10 μm, $n = 50$. X indicated that with the chosen microscopy settings, no signal was obtained. RAD51, ETO−, EV vs WT ($P = 0.6052$), EV vs ΔHAT ($P = 0.6391$). RAD51, ETO +, EV vs WT ($P < 0.0001$), EV vs ΔHAT ($P = 0.2167$). γH2AX, ETO−, EV vs WT ($P = 0.6194$), EV vs ΔHAT ($P = 0.7694$). γH2AX, ETO +, EV vs WT ($P < 0.0001$), EV vs ΔHAT ($P = 0.9586$). (F) Neutral comet assay (upper) and analysis of tail moment (lower) in HeLa cells transfected with indicated Flag-PCAF variants and treated with or without ETO (20 μM, 2 h). Scale bars, 100 μm. ETO−, EV vs WT ($P = 0.8539$), EV vs ΔHAT ($P = 0.5024$). ETO +, EV vs WT ($P < 0.0001$), EV vs ΔHAT ($P = 0.7454$). (G) Relative HR repair efficiency in U2OS cells transfected with indicated Flag-PCAF variants. EV vs WT ($P = 0.0004$), EV vs ΔHAT ($P = 0.9078$). (H, I) Cell survival assay in HeLa cells transfected with indicated Flag-PCAF variants in response to different doses of ETO (H) and ADR (I). ETO, 3.3 μM ($P = 000396$), 11 μM ($P = 0.005928$), 33 μM ($P = 0.013787$), 100 μM ($P = 0.582992$). ADR, 0.3 μM ($P < 0.0001$), 1.1 μM ($P < 0.0001$), 3.3 μM ($P < 0.0001$), 10 μM ($P = 0.030867$). All data are represented as mean ± SD of three independent experiments. $P$ values are from Student's $t$ tests (A–D, G–I) or Mann–Whitney $U$ test (E, F). ***$P < 0.001$, ns: not significant. Source data are available online for this figure.

down assay. Then take the cytoplasmic and chromatin fraction to pull down the RAD51 protein.

## Cell cycle analysis

HeLa cells were transfected with an empty vector, PCAF, or PCAF-ΔHAT for 48 h. The cells were then collected and washed twice with PBS. After washing, the cells were fixed in 70% ethanol at 4 °C for 12 h and subsequently stained with propidium iodide (PI) at 37 °C for 30 min. The cell cycle was analyzed using FACS (BD Biosciences, USA). Data were analyzed using ModFit.

## Protein purification and in vitro pull-down assays

*E.coli* (BL21 DE3 strain) transformed with His-RAD51 or His-PCAF was induced with 0.2 mM IPTG for 12 h at 16 °C. The His-tagged proteins were purified using the His-tag purification kit. After the centrifugation ($5000 \times g$ for 10 min at 4 °C), the pellet was suspended with the non-denaturing lysate, followed by sonication for 18 min (4 s for work and 6 s for intervals) and centrifugation to obtain the supernatant. Then the His-tagged proteins were affinity purified using the Ni-NTA agarose and washed five times with washing buffer. The bound proteins were eluted with elution buffer. Purified proteins were used for in vitro pull-down assays. His-RAD51 was incubated with RAD51 antibody, then immobilized to Protein A/G magnetic beads, which were then incubated with purified PCAF. After washing, the proteins were analyzed using western blot.

## In vitro acetylation assay

For the in vitro acetylation assays, 5 μg His-RAD51 and 5 μg His-PCAF were incubated in 50 μL acetyltransferase assay buffer (10% glycerol, 50 mM Tris-HCl, pH 8.0, 0.1 mM EDTA and dithiothreitol) with 20 μM Ac-CoA at 30 °C for 2 h. The SDS-loading buffer was used to stop the reaction. Then the proteins were separated by western blot.

## Electrophoretic mobility shift assay (EMSA)

The 60 bp FAM-labeled DNA probe (10 μM) was prepared in annealing buffer (10 mM Tris, 1 mM EDTA, 50 mM NaCl, pH 8.0)

and denatured at 95 °C for 10 min, followed by gradual cooling to 25 °C over 30 min to anneal. The binding reaction was performed in a 20 μL reaction buffer and incubated at 37 °C for 15 min. Samples were resolved on the 7% native polyacrylamide gel for 60 min. The DNA-protein complex was directly visualized using a fluorescence imaging system.

## Statistical analysis

Data were analyzed with GraphPad Prism 8.0 for statistical tests, such as Student's $t$ test or Mann–Whitney $U$ test, $P < 0.05$ indicates significant differences between groups, and ns indicates non-significant differences between groups. Image data were processed and analyzed using Photoshop and ImageJ. The data were statistically analyzed using GraphPad Prism. All experimental data were obtained from three independent experiments.

# Data availability

Our study includes no data deposited in public repositories.

The source data of this paper are collected in the following database record: biostudies:S-SCDT-10_1038-S44319-025-00513-6.

# Peer review information

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

## Acknowledgements

The authors thank Dr. Songbai Liu (Suzhou Vocational Health College, Suzhou, China) for providing the U2OS DR-EGFP cell line. This work was supported by the National Natural Science Foundation of China (grant 82373183) and Natural Science Research of Jiangsu Higher Education Institutions of China (grant 23KJA180002).

## Author contributions

**Jiajia Hou**: Conceptualization; Resources; Data curation; Software; Formal analysis; Supervision; Validation; Investigation; Visualization; Methodology; Writing—original draft; Project administration; Writing—review and editing. **Munan Shi**: Conceptualization; Resources; Data curation; Software; Formal analysis; Supervision; Validation; Visualization; Methodology; Writing—original draft; Writing—review and editing. **Jialu Hong**: Conceptualization; Resources; Data curation; Software; Formal analysis; Supervision; Validation. **Yuting Liu**: Software; Formal analysis; Supervision; Validation; Visualization. **Xinyi Song**: Software; Supervision; Validation; Visualization. **Haipeng Rao**: Supervision; Validation; Visualization; Methodology. **Ying Ma**: Resources; Software; Supervision; Validation. **Chunchun Huang**: Software; Formal analysis; Supervision; Visualization. **Zhigang Hu**: Supervision. **Lingfeng He**: Supervision. **Zhigang Guo**: Conceptualization; Resources; Supervision; Funding acquisition; Validation; Visualization; Project administration. **Feiyan Pan**: Conceptualization; Resources; Supervision; Funding acquisition; Visualization; Methodology; Writing—original draft; Project administration; Writing—review and editing.

Source data underlying figure panels in this paper may have individual authorship assigned. Where available, figure panel/source data authorship is listed in the following database record: biostudies:S-SCDT-10_1038-S44319-025-00513-6.

## Disclosure and competing interests statement

The authors declare no competing interests.

# Expanded View Figures

**A**

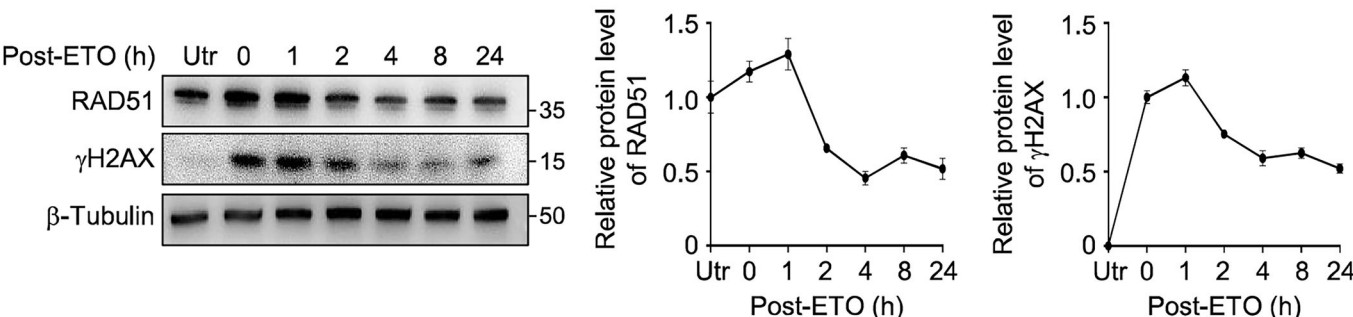

**B**

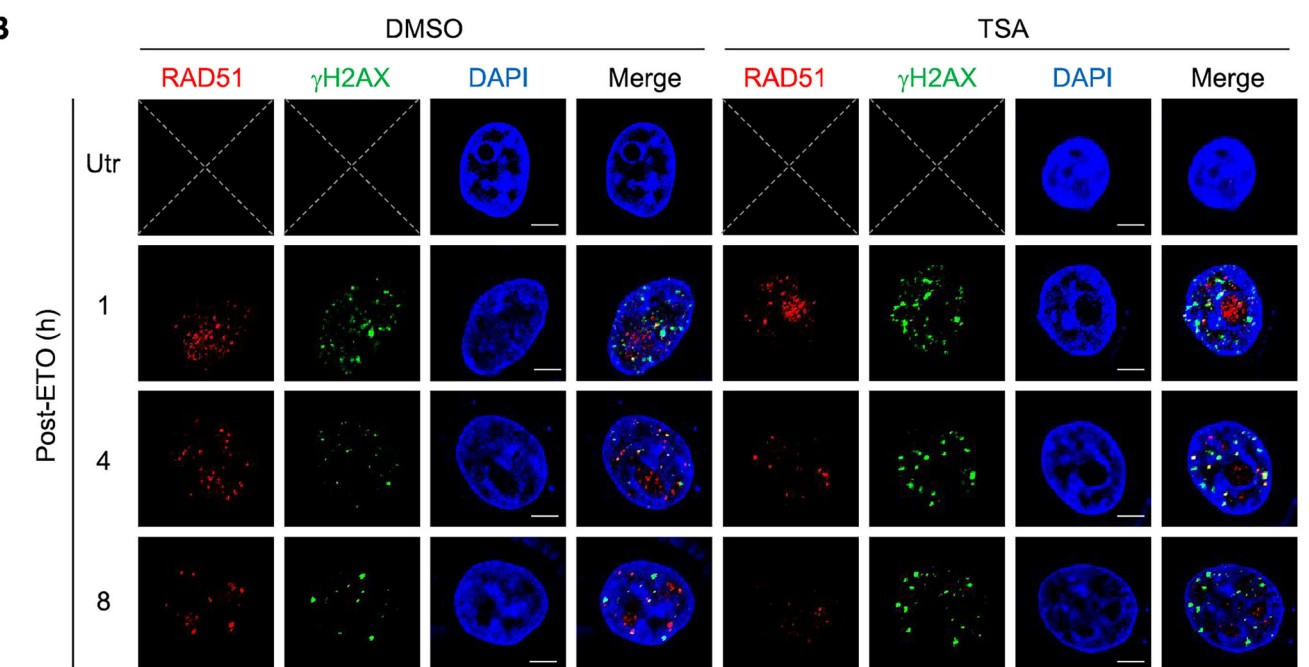

**Figure EV1. RAD51 protein levels change dynamically during HR.**

(A) Immunoblot of RAD51 and γH2AX in HeLa cells untreated or treated with 20 μM ETO for 2 h and recovered at the indicated time points (left). Quantifications of RAD51 and γH2AX are shown on the right. Data represented as mean ± SD of three independent experiments. (B) Representative immunofluorescence images of RAD51 (red) and γH2AX (green) foci in HeLa cells pretreated with DMSO and TSA (0.5 μM, 6 h), followed by ETO exposure (20 μM, 2 h), with cell lysates recovered at the indicated time points. DNA was stained by DAPI (blue). Scale bars, 10 μm. X indicated that with the chosen microscopy settings, no signal was obtained. Source data are available online for this figure.

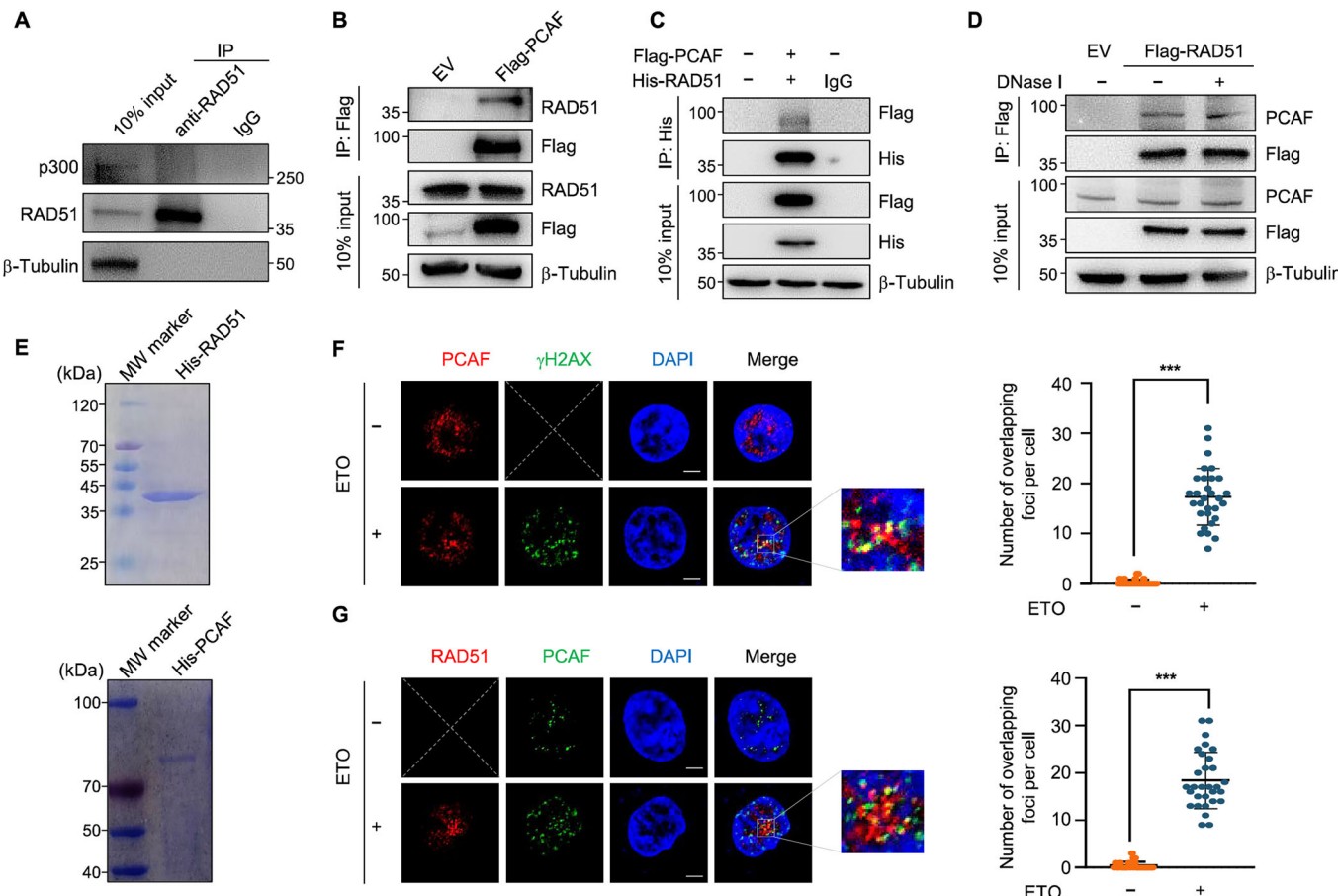

**Figure EV2. PCAF interacts with RAD51.**

(A) Immunoprecipitation to detect the interaction between RAD51 and p300 in HEK293T cells. (B) Immunoblot of RAD51 in anti-Flag immunoprecipitates from HEK293T cells transfected with empty vector or Flag-PCAF. (C) Immunoblot of Flag-PCAF in anti-His immunoprecipitates from HEK293T cells co-transfected with Flag-PCAF and His-RAD51 or empty vector. (D) Immunoblot of PCAF in anti-Flag-RAD51 immunoprecipitates from HEK293T cells transfected with empty vector or Flag-RAD51 and treated with or without DNase I. (E) Coomassie blue staining of purified His-RAD51 (upper) and His-PCAF (lower) protein. (F) Representative immunofluorescence images and quantification ($n = 30$) of PCAF (red) and γH2AX (green) foci in HeLa cells treated with or without ETO (20 μM, 2 h) and recovered for 1 h. Scale bars, 10 μm. X indicated that with the chosen microscopy settings, no signal was obtained. $P < 0.0001$. (G) Representative immunofluorescence images and quantification ($n = 30$) of RAD51 (red) and PCAF (green) foci in HeLa cells treated with or without ETO (20 μM, 2 h) and recovered for 1 h. Scale bars, 10 μm. X indicated that with the chosen microscopy settings, no signal was obtained. $P < 0.0001$. All data are represented as mean ± SD of three independent experiments. $P$ values are from Mann–Whitney $U$ test (F, G). ***$P < 0.001$. Source data are available online for this figure.

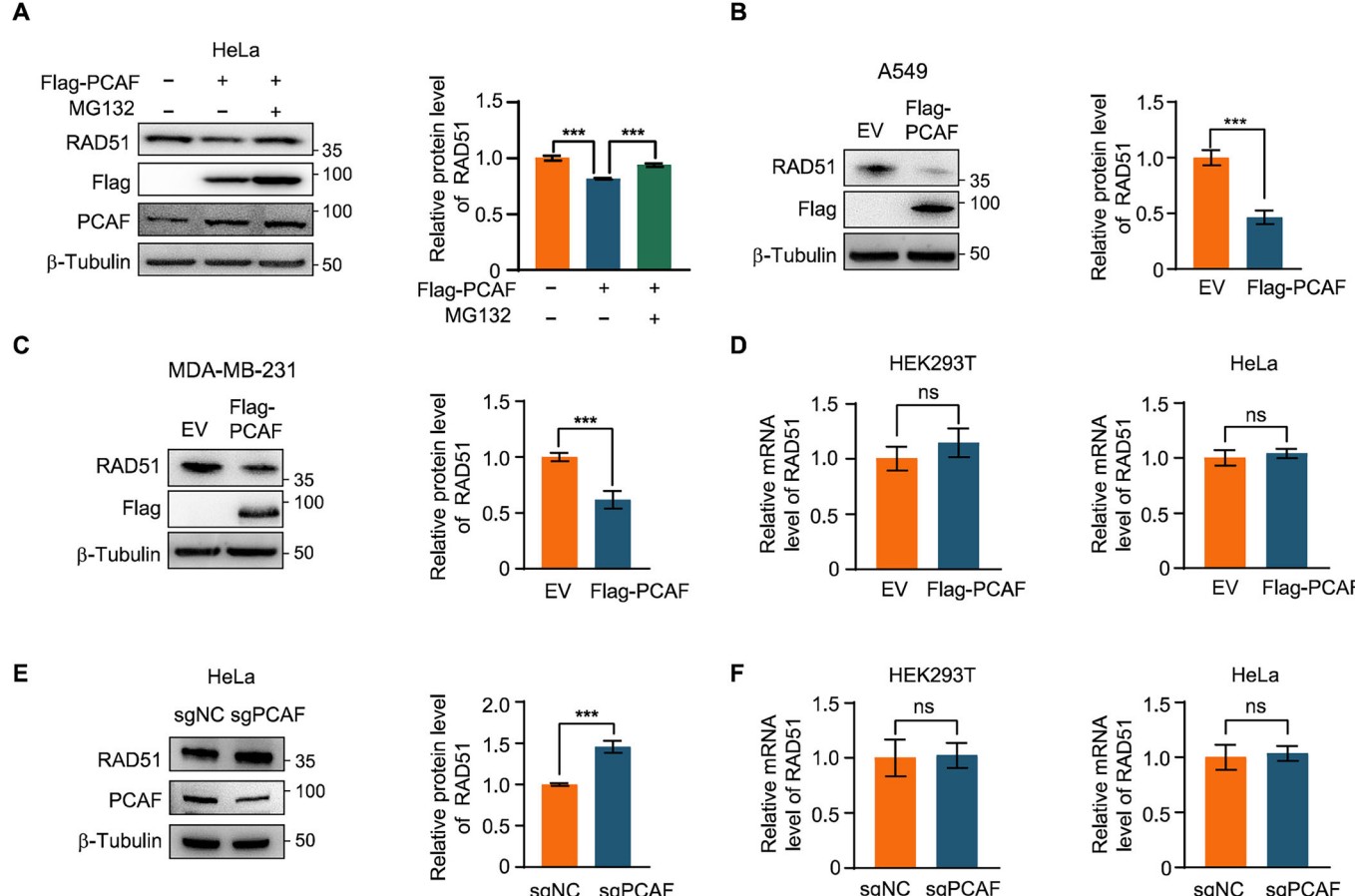

**Figure EV3. PCAF does not affect the transcription of RAD51.**

(A) Immunoblot (left) and quantification (right) of RAD51 in HeLa cells transfected with or without Flag-PCAF first, then treated with or without 10 μM MG132 for 6 h. EV vs Flag-PCAF ($P = 0.0002$), Flag-PCAF vs Flag-PCAF + MG132 ($P = 0.0002$). (B, C) Immunoblot (left) and quantification (right) of RAD51 in A549 (B) and MDA-MB-231 (C) cells transfected with or without Flag-PCAF. (B) ($P = 0.0006$), (C) ($P = 0.00097$). (D) Relative mRNA level of RAD51 in HEK293T (left) and HeLa (right) cells transfected with empty vector or Flag-PCAF for 48 h. HEK293T ($P = 0.7592$), HeLa ($P = 0.5225$). (E) Immunoblot (left) and quantification (right) of RAD51 in HeLa cells transfected with sgNC or sgPCAF. $P = 0.0004$. (F) Relative mRNA level of RAD51 in HEK293T (left) and HeLa (right) cells transfected with sgNC or sgPCAF. HEK293T ($P = 0.6320$), HeLa ($P = 0.5095$). All data are represented as mean ± SD of three independent experiments. $P$ values are from Student's $t$ tests. \*\*\*$P < 0.001$; ns not significant (A–F).

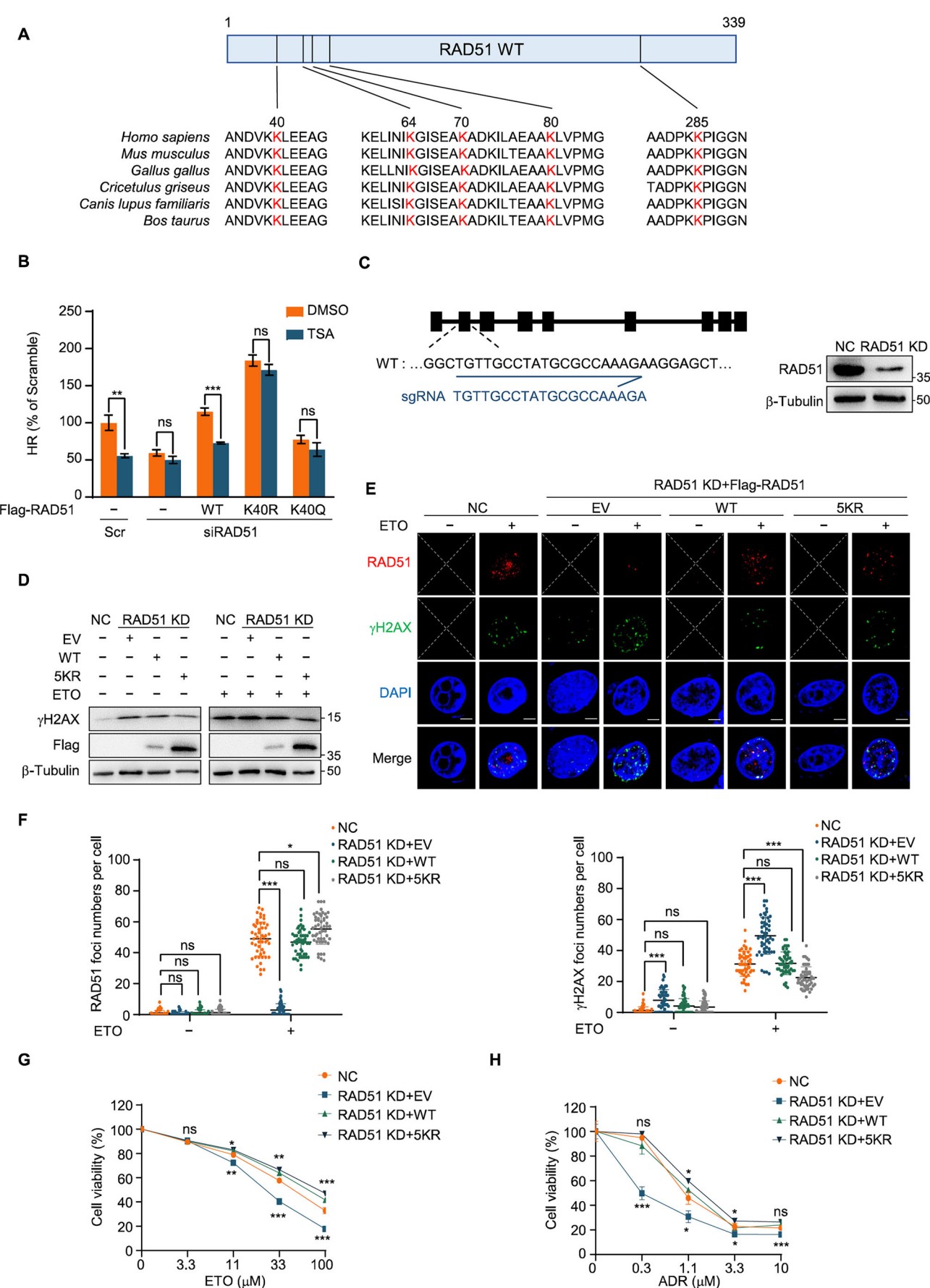

◀ **Figure EV4. RAD51 acetylation regulates HR.**

(A) Diagram showing the sequence of acetylation sites of RAD51. (B) HR efficiency in U2OS cells transfected with scramble RNA or siRAD51 for 24 h, followed by transfection with the indicated Flag-RAD51 mutations for another 24 h, and treated with DMSO and TSA (0.5 μM, 6 h). Scr ($P = 0.001898$), siRAD51 ($P = 0.066415$), WT ($P = 0.000142$), K40R ($P = 0.103938$), K40Q ($P = 0.096139$). (C) Schematic diagram of CRISPR/Cas9 targeting RAD51 (left) and immunoblot (right) of RAD51 in RAD51 KD HeLa cells. (D) Immunoblot of γH2AX in RAD51 KD HeLa cells transfected with indicated Flag-RAD51 mutations and treated with or without ETO (20 μM, 2 h). (E) Representative immunofluorescence images of RAD51 (red) and γH2AX (green) foci in RAD51 KD HeLa cells transfected with different Flag-RAD51 mutations and treated with or without ETO (20 μM, 2 h). DNA was stained by DAPI (blue). Scale bars, 10 μm. X indicated that with the chosen microscopy settings, no signal was obtained. (F) Quantification of RAD51 (right) or γH2AX (left) foci per cell from (E) ($n = 50$). RAD51, ETO −, NC vs RAD51 KD + EV ($P = 0.1333$), NC vs RAD51 KD + WT ($P = 0.7142$), NC vs RAD51 KD + 5KR ($P = 0.4249$). RAD51, ETO +, NC vs RAD51 KD + EV ($P < 0.0001$), NC vs RAD51 KD + WT ($P = 0.2425$), NC vs RAD51 KD + 5KR ($P = 0.0013$). γH2AX, ETO −, NC vs RAD51 KD + EV ($P < 0.0001$), NC vs RAD51 KD + WT ($P = 0.3234$), NC vs RAD51 KD + 5KR ($P = 0.165271$). γH2AX, ETO +, NC vs RAD51 KD + EV ($P < 0.0001$), NC vs RAD51 KD + WT ($P = 0.8152$), NC vs RAD51 KD + 5KR ($P < 0.0001$). (G, H) Cell survival assay was performed in RAD51 KD HeLa cells transfected with indicated Flag-RAD51 mutations in response to different doses of ETO (G) and ADR (H). ETO, NC vs RAD51 KD + 5KR, 3.3 μM ($P = 0.237876$), 11 μM ($P = 0.021469$), 33 μM ($P = 0.001285$), 100 μM ($P = 0.00063$). NC vs RAD51 KD + EV, 3.3 μM ($P = 0.58183$), 11 μM ($P = 0.005927$), 33 μM ($P = 0.00068$), 100 μM ($P = 0.000489$). ADR, NC vs RAD51 KD + 5KR, 0.3 μM ($P = 0.020424$), 1.1 μM ($P = 0.008277$), 3.3 μM ($P = 0.060957$), 10 μM ($P = 0.002279$). NC vs RAD51 KD + EV, 0.3 μM ($P = 0.000134$), 1.1 μM ($P = 0.019614$), 3.3 μM ($P = 0.016448$), 10 μM ($P = 0.000114$). All data are represented as mean ± SD of three independent experiments. $P$ values are from Mann–Whitney $U$ test (F) or Student's $t$ tests (B, G, H). *$P < 0.05$, **$P < 0.01$, ***$P < 0.001$, ns not significant. Source data are available online for this figure.

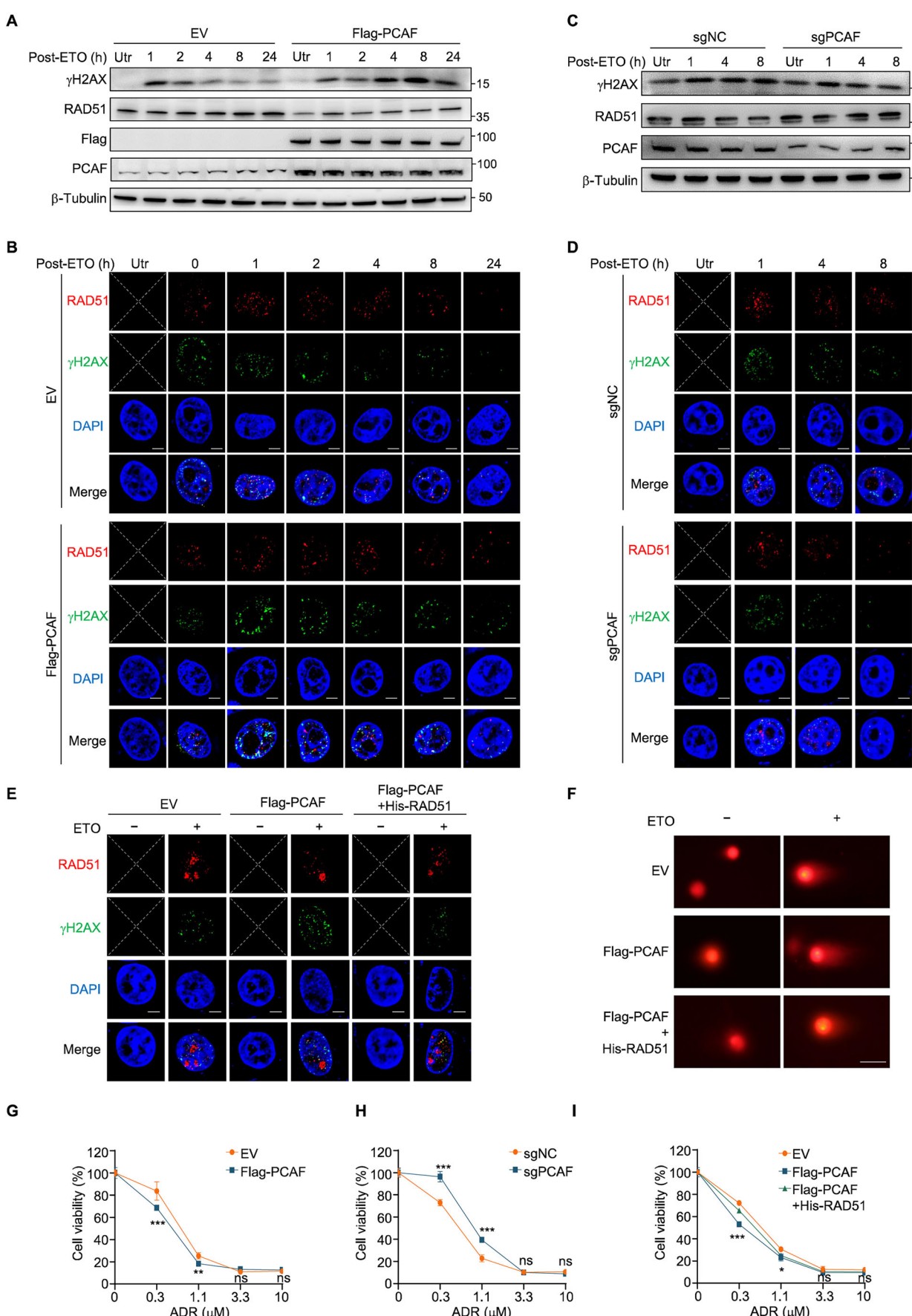

◀ **Figure EV5. PCAF is involved in HR.**

(A) Immunoblot of γH2AX and RAD51 in HeLa cells transfected with empty vector or Flag-PCAF, treated with 20 μM ETO for 2 h and recovered at the indicated time points. (B) Representative immunofluorescence images of RAD51 (red) and γH2AX (green) foci in HeLa cells transfected with empty vector or Flag-PCAF, followed by ETO exposure (20 μM, 2 h), with cell lysates recovered at the indicated time points. DNA was stained by DAPI (blue). Scale bars, 10 μm. X indicated that with the chosen microscopy settings, no signal was obtained. (C) Immunoblot of γH2AX and RAD51 in HeLa cells transfected with sgNC or sgPCAF, treated with 20 μM ETO for 2 h and recovered at the indicated time points. (D) Representative immunofluorescence images of RAD51 (red) and γH2AX (green) foci in HeLa cells transfected with sgNC or sgPCAF, followed by ETO exposure (20 μM, 2 h), with cell lysates recovered at the indicated time points. DNA was stained by DAPI (blue). Scale bars, 10 μm. X indicated that with the chosen microscopy settings, no signal was obtained. (E) Representative immunofluorescence images of RAD51 (red) and γH2AX (green) foci in HeLa cells transfected with Flag-PCAF or Flag-PCAF and His-RAD51, followed by ETO exposure (20 μM, 2 h), with cell lysates recovered at the indicated time points. DNA was stained by DAPI (blue). Scale bars, 10 μm. X indicated that with the chosen microscopy settings, no signal was obtained. (F) Neutral comet assay in HeLa cells overexpressing PCAF or co-transfected with Flag-PCAF and His-RAD51, with or without ETO treatment (20 μM, 2 h). Scale bars, 100 μm. X indicated that with the chosen microscopy settings, no signal was obtained. (G) Cell survival assay was performed in HeLa cells transfected with empty vector or Flag-PCAF in response to different doses of ADR. 0.3 μM ($P < 0.0001$), 1.1 μM ($P = 0.00228$), 3.3 μM ($P = 0.287791$), 10 μM ($P = 0.522845$). (H) Cell survival assay was performed in HeLa cells transfected with sgNC or sgPCAF in response to different doses of ADR. 0.3 μM ($P = 0.000153$), 1.1 μM ($P = 0.000848$), 3.3 μM ($P = 0.371428$), 10 μM ($P = 0.24121$). (I) Cell survival assay was performed in HeLa cells transfected with PCAF or co-transfected with Flag-PCAF and His-RAD51 in response to different doses of ADR. EV vs Flag-PCAF, 0.3 μM ($P = 0.000107$), 1.1 μM ($P = 0.029975$), 3.3 μM ($P = 0.192603$), 10 μM ($P = 0.166067$). All data are represented as mean ± SD of three independent experiments. *P* values are from Student's *t* tests (G–I). **$P < 0.01$, ***$P < 0.001$, ns: not significant. Source data are available online for this figure.

