## [Peer Review File · EMBO Reports]

PCAF-Mediated Acetylation Regulates RAD51 Dynamic Localization on Chromatin during HR Repair

Jiajia Hou, Munan Shi, Jialu Hong, Yuting Liu, Xinyi Song, Haipeng Rao, Ying Ma, Chunchun Huang, Zhigang Hu, Lingfeng He, Zhigang Guo, and Feiyan Pan

Corresponding author(s): Feiyan Pan (panfeiyang@njnu.edu.cn) , Zhigang Guo (guo@njnu.edu.cn)

Review Timeline:

Submission Date:	5th Nov 24
Editorial Decision:	10th Dec 24
Revision Received:	19th Mar 25
Editorial Decision:	22nd Apr 25
Revision Received:	19th May 25
Editorial Decision:	5th Jun 25
Revision Received:	9th Jun 25
Accepted:	17th Jun 25

Editor: Esther Schnapp

Transaction Report:

Dear Dr. Pan,

Thank you for the submission of your manuscript to EMBO reports. We have now received the full set of referee reports that is pasted below.

As you will see, the referees acknowledge that the findings are potentially interesting. However, all of them also point out that significant revisions will be required to strengthen the study, to place it better into context, and to uncover how the DDR regulates PCAF-controlled RAD51 levels. I think all referee comments are good and should be addressed. Please let me know in case you disagree and we can discuss the exact revision requirements further, also in a video chat, if you like.

I would thus like to invite you to revise your manuscript with the understanding that the referee concerns must be fully addressed and their suggestions taken on board. Please address all referee concerns in a complete point-by-point response. Acceptance of the manuscript will depend on a positive outcome of a second round of review. It is EMBO reports policy to allow a single round of major revision only and acceptance or rejection of the manuscript will therefore depend on the completeness of your responses included in the next, final version of the manuscript.

We realize that it is difficult to revise to a specific deadline. In the interest of protecting the conceptual advance provided by the work, we recommend a revision within 3 months (12th Mar 2025). Please discuss the revision progress ahead of this time with the editor if you require more time to complete the revisions.

- 1) A data availability section providing access to data deposited in public databases is missing. If you have not deposited any data, please add a sentence to the data availability section that explains that.
- 2) Your manuscript contains statistics and error bars based on $n=2$. Please use scatter blots in these cases. No statistics should be calculated if $n=2$.

3) We replaced Supplementary Information with Expanded View (EV) Figures and Tables that are collapsible/expandable online. A maximum of 5 EV Figures can be typeset. EV Figures should be cited as 'Figure EV1, Figure EV2' etc... in the text and their respective legends should be included in the main text after the legends of regular figures.

5) a complete author checklist, which you can download from our author guidelines <https://www.embopress.org/page/journal/14693178/authorguide>. Please insert information in the checklist that is also reflected in the manuscript. The completed author checklist will also be part of the RPF.

6) Please note that all corresponding authors are required to supply an ORCID ID for their name upon submission of a revised manuscript (<https://orcid.org/>). Please find instructions on how to link your ORCID ID to your account in our manuscript tracking system in our Author guidelines <https://www.embopress.org/page/journal/14693178/authorguide#authorshipguidelines>

10) Regarding data quantification (see Figure Legends:

<https://www.embopress.org/page/journal/14693178/authorguide#figureformat>)

12) All Materials and Methods need to be described in the main text using our 'Structured Methods' format, which is required for all research articles. According to this format, the Methods section includes a separate Reagents and Tools Table file (listing key reagents, experimental models, software and relevant equipment and including their sources and relevant identifiers) and a Methods and Protocols section describing the methods using a step-by-step protocol format. The aim is to facilitate adoption of the methodologies across labs. More information on how to adhere to this format as well as a downloadable template (.docx) for the Reagents and Tools Table can be found in our author guidelines:

An example of a Method paper with Structured Methods can be found here: <https://www.embopress.org/doi/full/10.1038/s44320-024-00037-6#sec-4>

I look forward to seeing a revised form of your manuscript when it is ready.

Yours sincerely,

Referee #1:

Hou, et. al. examined the role of protein acetylation on RAD51 degradation. They previously reported that HDAC1 deacetylates RAD51 and this leads to HR promotion (Shi et al, 2023, EMBO J). In this new study, they have sought to understand the mechanism of RAD51 acetylation and its consequences. They demonstrate by cell biology that acetylated RAD51 increases after DNA damage induction, peaking at two hours before tapering off, and that Zn²⁺ dependent histone deacetylases (HDACs) appear to be responsible for the acetylation. The authors saw HR impairment upon inhibition of HDACs. After testing several lysine acetyltransferases (KATs), the authors discovered PCAF as an important RAD51 KAT and they showed that the acetyltransferase domain of PCAF appears to mediate RAD51 interaction. Ectopic overexpression of PCAF suggests that acetylation of RAD51 causes degradation of RAD51 by the ubiquitin-proteasome pathway. They then identified K40 as a major site and introduction of mutations that mimic acetylation or deacetylation led to increased and decreased degradation of RAD51 respectively. Based on these findings, they present a mechanism whereby PCAF acetylates RAD51 at K40 to trigger RAD51 degradation by the ubiquitin-proteasome system.

The results are interesting and of general interests. However, there are significant issues that need to be addressed.

Specific comments:

1. Identification of RAD51 acetylation: In this study, all of the detection of acetylated RAD51 (ace-RAD51) was performed using an anti-acetylated lysine antibody, which is not specific to RAD51. Consider providing additional evidence for ace-RAD51 levels using a different approach.
2. Fig. 1A, 1C: RAD51 IP panels: The RAD51 IP panels appear to be reblotted images. Is this the case? Western blot images obtained from separate blots are preferred.
3. RAD51 knockout cells (Fig. 4. D-G, Fig. S4.): RAD51 is essential for cell viability of human cells, thus complete RAD51 knockout is likely lethal. The authors should rigorously verify the authenticity of their RAD51 knockout cell line - sequencing the cell line, measuring HR efficiency, cell cycle profile etc.
4. Endogenous PCAF: At several points in the study, the endogenous PCAF level is not presented, making it challenging to interpret the PCAF-related phenotypes. Please include Western blot data showing PCAF levels (e.g., Fig. 3B, 3F, Fig. S3A, and Fig. S5A).
5. Fig. 3G, H, Ubi panel (line 169): The statement, "both RAD51 acetylation and ubiquitin binding to RAD51 were increased in PCAF overexpression cells," requires clarification by providing direct evidence showing that the Ubi bands are ubiquitinated RAD51 species.
6. Fig. S2B: Either the label 'Flag-PCAF1' in the figure or the mention of 'anti-FLAG-RAD51' in the figure legend is incorrect.
7. At several points in the manuscript, quantification of results would significantly strengthen the claims regarding protein levels, e.g. in Fig. 1B and Fig. 6E. Explain how loading normalization and result quantification were conducted.
8. Fig. 1D: Cycloheximide was included in the experiment, but its use is not described until later in the paper.
9. Fig. 1E: The description of the experiment is unclear, particularly regarding TSA treatment. The methods section does not provide sufficient detail about the conditions or protocol used.
10. Fig. 3C: Include a description in the methods how knockdown was achieved.

Referee #2:

In this study, Hou and colleagues aim to advance upon their previous identification of Rad51 regulation by acetylation. This is motivated by their previous observation that HDAC1 can promote homologous recombination DNA repair and the de-acetylation of Rad51. The authors identify PCAF as the primary acetyltransferase responsible for Rad51 acetylation, acting on Lys40 on Rad51. They go on to provide evidence that this acetylation promotes the destabilization of Rad51 protein through proteasomal degradation. The work presented here, while conceptually interesting, suffers from poorly controlled experiments and fails to meaningfully integrate these new findings with the already well characterized roles of PCAF and Rad51 in DNA repair. Many results presented here are not entirely novel, including the effect of HDAC inhibition on HR and the identification of the acetylation site on Rad51. Several controls for essential findings are absent and validation for the experimental conditions used

is not presented, making it difficult to assess the rigor and impact of the findings. Specific comments are listed below.

MAJOR:

1. PCAF regulation of HR is well known, importantly, this previous work has identified PCAF expression as a positive regulator of HR, not a suppressor of this repair pathway. For the results presented here to be of value to the research community some effort is needed to place these new findings within the context of what is known about PCAF and DNA repair, both experimentally and in the text as this was completely absent from this manuscript.
2. A citation should be included for PMID: 31753913, the study which first described a HR repair defect in PCAF depleted cells. Additionally, the citation for PMID: 32966758 on line 76 is incorrect, this study identified a role for PCAF in replication fork degradation and is not appropriate for the transcriptional roles of PCAF, which were described years if not decades prior.
3. Are levels of H2BK120ac changed in the cell lines and conditions used here given the established role of PCAF in this process?
4. It is surprising that the authors do not include any work building on their previous studies regarding regulation of HR and Rad51 levels by GAPDH, demonstrating that GAPDH KD works synergistically with PCAF overexpression for Rad51 levels would strengthen both of those claims.
5. Does PCAF overexpression cancer types also present with decreased Rad51 levels? Is the pathway described here relevant in cancer models?
6. Inhibition of DNA repair and sensitivity to DSBs by TSA treatment is well known (PMID: 17722998).
7. Is the interaction between PCAF and Rad51 enhanced by DNA damage? All of the interaction studies presented in figure 2 appear to be in non-damaged conditions, which may be less relevant to the pathway described.
8. Figures 3B and 6B do not include a blot for endogenous PCAF levels, the strength of these results is difficult to determine without knowing the expression level of FLAG-PCAF relative to wild type. These experiments should be done side by side, the level of PCAF overexpression needed to achieve Rad51 loss and HR suppression is critical to understand the impact of this claim.
9. It is concerning that the Rad51 K40Q mutant partially reduces γ H2AX foci in Rad51 KO despite lower levels of Rad51 foci formation. It is also concerning that a Rad51 KO is used in this study, as it has been previously identified as an essential gene for cell proliferation (PMID: 33095861). No blot or qPCR is provided to validate this cell line.
10. The role of RAD51 acetylation in regulating DNA repair is not very clear. Although phenotypes are correlated, how does acetylation of RAD51 promote these activities molecularly and mechanistically? As is, this is more observational than providing mechanistic understanding for these findings. For example, does acetylation of RAD51 play no role in HR repair? Is overexpression of PCAF then just a detrimental consequence of these conditions? It seems more likely that this acetylation is regulatory and plays roles in normal repair but when dysregulated, because detrimental to repair. A clearer message from the data provided needs to be synthesized into a molecular and mechanistic model to understand the relevance and impact of these findings.

MINOR

1. An un-damaged control would be useful in figure 1B, to demonstrate that Rad51 chromatin association is increased after damage.
2. In the blots presented in figure 3 E and F the changes in Rad51 levels look visually less than the accompanying quantification. Lower exposures for these blots might better highlight the differences in protein levels.

Referee #3:

In this manuscript, Hou and colleagues performed several experiments to examine the mechanism and the function of RAD51 acetylation during DSB repair. They show that RAD51 interacts with and is acetylated by PCAF. They map the interaction domains between both proteins, identify a key residue that is likely acetylated by PCAF and suggest that RAD51 acetylation by PCAF promotes its proteasome-mediated degradation. Lastly, they demonstrate a requirement for the acetylation domain of PCAF and suggest that acetylation of RAD51 by PCAF downregulates homologous recombination (HR) by increasing RAD51 turnover.

Overall, the work is interesting and increases our understanding of the modifications regulating the stability, localization and activity of RAD51. The data is also largely well-presented, and the questions have been extensively investigated. However, the main problem with the work is that some key data show relatively minor differences that, while significant in some cases, may not be critical for RAD51 and HR regulation especially since a lot of the phenotypes stem from PCAF overexpression. This weakness feeds into another problem in which there are quite a few instances where the data have been overinterpreted. To strengthen the manuscript, an overarching mechanism for how this potentially key role of PCAF in controlling RAD51 levels is regulated by DDR and how it integrates with known mechanisms of RAD51 degradation like RFW3 is missing or inadequate and ought to be addressed.

Concerns

The changes in Fig 1B are not readily evident from the blot and needs to be quantified.

While TSA treatment reduces HR in Figure 1E, that is not necessarily related to Rad51. The authors need to repeat this experiment in a later figure showing that the RAD51 acetylation mutant doesn't show diminished HR upon TSA treatment.

How is this pathway regulated and controlled by DDR? In Fig 2E, we already see significant interaction in the absence of drug and is definitely not 'greatly enhanced' (line 131). The 'increase' appears to be due to reduced RAD51 levels in the input, PCAF levels appear unchanged. Also, 2E is better interpreted with endogenous PCAF IP'ed with RAD51.

Does PCAF colocalize with RAD51? A quick IF can address this.

Figure 4G shows very weak differences that is not consistent with the interpretation. The only clearcut differences are with RAD51 KO.

Indices of significance are lacking in Figure 5A

Figure 5 will be strengthened by performing the PCAF KD using the HR reporter.

In Fig 5, at later time points, 8 hrs and 24 hrs post etoposide treatment, there is a significantly higher number of RAD51 foci in PCAF overexpressing cells compared to EV control. Similarly, PCAF knockdown cells have fewer RAD51 foci than sgNC cells 4 hrs and 8 hrs post Etoposide treatment. How do the authors reconcile this?

What is going on in Fig 6C? The figure is not consistent with the quantification. Based on the blot shown, the level of ubiquitination in the cells transfected with EV is similar to the PCAF-WT OE.

More importantly, RAD51's size is unchanged in the ubiquitinated blot (the most prominent band is at 37KD even though ubiquitin is 8.5KD). This experiment (and the ones in Figure 3G and H) are not convincing as indicative of RAD51 ubiquitination and is better done in well controlled in vitro conditions.

In Fig 6H, the authors should examine the cell cycle as a control.

Is knockdown of RFWD3 additive or synergistic with sgRNA to PCAF?

Minor

Title of Supp 1 is misleading (they're not looking at acetylation)

Line 100 is a stretch. There's no link as of yet between acetylation and stability. The authors can say something like we wanted to investigate the possibility.

The figures referenced in discussion should be cited.

Referee #1:

Hou, et. al. examined the role of protein acetylation on RAD51 degradation. They previously reported that HDAC1 deacetylates RAD51 and this leads to HR promotion (Shi et al, 2023, EMBO J). In this new study, they have sought to understand the mechanism of RAD51 acetylation and its consequences. They demonstrate by cell biology that acetylated RAD51 increases after DNA damage induction, peaking at two hours before tapering off, and that Zn²⁺ dependent histone deacetylases (HDACs) appear to be responsible for the acetylation. The authors saw HR impairment upon inhibition of HDACs. After testing several lysine acetyltransferases (KATs), the authors discovered PCAF as an important RAD51 KAT and they showed that the acetyltransferase domain of PCAF appears to mediate RAD51 interaction. Ectopic overexpression of PCAF suggests that acetylation of RAD51 causes degradation of RAD51 by the ubiquitin-proteasome pathway. They then identified K40 as a major site and introduction of mutations that mimic acetylation or deacetylation led to increased and decreased degradation of RAD51 respectively. Based on these findings, they present a mechanism whereby PCAF acetylates RAD51 at K40 to trigger RAD51 degradation by the ubiquitin-proteasome system.

The results are interesting and of general interests. However, there are significant issues that need to be addressed.

Response: We sincerely appreciate the time and effort the reviewer invested in reviewing our manuscript and providing thoughtful comments. Based on your suggestions, we have conducted the necessary experiments and revised the manuscript to address your concerns. We believe the revisions have greatly enhanced the quality of the manuscript.

Specific comments:

1. Identification of RAD51 acetylation: In this study, all of the detection of acetylated RAD51 (ace-RAD51) was performed using an anti-acetylated lysine antibody, which is not specific to RAD51. Consider providing additional evidence for ace-RAD51 levels using a different approach.

Response: We understand the reviewer's concern. In our previous study, we performed the acetylation of purified RAD51 in vitro using HeLa whole cell lysate as an acetyltransferase donor and identified five acetylation sites (lysine 40, 64, 70, 80, and 285) by mass spectrometry, which provided primary evidence that RAD51 undergoes acetylation modification.

Following your suggestion, we purified PCAF and subsequently re-conducted the in vitro acetylation assay of RAD51 using the purified protein. Our results further confirmed the acetylation of RAD51 and also demonstrated that PCAF is the acetyltransferase for RAD51 acetylation (Figure 2D).

Figure for referees not shown.

2. Fig. 1A, 1C: RAD51 IP panels: The RAD51 IP panels appear to be reblotted images. Is this the case? Western blot images obtained from separate blots are preferred.

Response: We thank the reviewer for the expert insights. Yes, the acetylated RAD51 was re-blotted with pan-acetylated antibodies after stripping. We have re-performed the ace-lysine western blot and updated the corresponding figure in the revised manuscript (Figure 1A and 1D).

3. RAD51 knockout cells (Fig. 4. D-G, Fig. S4.): RAD51 is essential for cell viability of human cells, thus complete RAD51 knockout is likely lethal. The authors should rigorously verify the authenticity of their RAD51 knockout cell line - sequencing the cell line, measuring HR efficiency, cell cycle profile etc.

Response: Thank you for bringing this to our attention. We utilized the CRISPR/Cas9 system followed by Blasticidin S selection to knockout RAD51. After drug screening, Western blot analysis of the surviving cells revealed that RAD51 was nearly undetectable under short exposure, leading us to designate these cells as "RAD51 KO" cells. Based on your recommendation, we extracted genomic DNA from these cells and performed sequencing, which indicated that the cell population was heterogeneous. Furthermore, upon repeating the Western blot with extended exposure, we detected low levels of RAD51 protein (Figure EV4C). These findings suggest that the RAD51 KO cells were not true knockout but rather knockdown (KD) cells. Accordingly, we have revised the relevant descriptions in the manuscript, replacing "KO" with "KD".

4. Endogenous PCAF: At several points in the study, the endogenous PCAF level is not presented, making it challenging to interpret the PCAF-related phenotypes. Please include Western blot data showing PCAF levels (e.g., Fig. 3B, 3F, Fig. S3A, and Fig. S5A).

Response: We fully agree with the reviewer that distinguishing endogenous and exogenous proteins is essential. However, due to the minimal size difference (1 kDa) between exogenous

Flag-PCAF and endogenous PCAF, incubation with the anti-PCAF antibody resulted in the detection of a single band across all groups, with a significantly increased intensity in the overexpression group. To more clearly show the endogenous and exogenous levels of PCAF, we have now included the Western blot data using both anti-PCAF and anti-Flag antibodies in the revised version (Figure 3B, 3F, EV3A and EV5A).

5. Fig. 3G, H, Ubi panel (line 169): The statement, "both RAD51 acetylation and ubiquitin binding to RAD51 were increased in PCAF overexpression cells," requires clarification by providing direct evidence showing that the Ubi bands are ubiquitinated RAD51 species.

Response: As the reviewer suggested, we have co-transfected with the Flag-PCAF or sgPCAF along with His-Ubi in HEK293T cells. Following His-Ubi pull-down assays, we detected RAD51 ubiquitination levels by Western blot analysis. A clear Western blot image has been included in the revised version (Figure 3G and 3J).

6. Fig. S2B: Either the label 'Flag-PCAF' in the figure or the mention of 'anti-FLAG-RAD51' in the figure legend is incorrect.

Response: We thank the reviewer for pointing out this typo. This figure label has been corrected accordingly in the revised manuscript (Figure EV2D).

7. At several points in the manuscript, quantification of results would significantly strengthen the claims regarding protein levels, e.g. in Fig. 1B and Fig. 6E. Explain how loading normalization and result quantification were conducted.

Response: Thank you for your suggestion. We have added the quantification of relative protein levels in Figure 1C and Appendix Figure S1E.

8. Fig. 1D: Cycloheximide was included in the experiment, but its use is not described until later in the paper.

Response: Thank you for pointing out the issue. We have revised the description of the cycloheximide in the article (Line 106-108).

9. Fig. 1E: The description of the experiment is unclear, particularly regarding TSA treatment. The methods section does not provide sufficient detail about the conditions or protocol used.

Response: Thank you. The Materials and Methods section has been revised and further detailed in the updated manuscript (Line 366-373).

10. Fig. 3C: Include a description in the methods how knockdown was achieved.

Response: Thank you. The detailed description of the knockdown methodology has been added to the materials and methods section (Line 355-359).

Referee #2:

In this study, Hou and colleagues aim to advance upon their previous identification of Rad51 regulation by acetylation. This is motivated by their previous observation that HDAC1 can

promote homologous recombination DNA repair and the de-acetylation of Rad51. The authors identify PCAF as the primary acetyltransferase responsible for Rad51 acetylation, acting on Lys40 on Rad51. They go on to provide evidence that this acetylation promotes the destabilization of Rad51 protein through proteasomal degradation. The work presented here, while conceptually interesting, suffers from poorly controlled experiments and fails to meaningfully integrate these new findings with the already well characterized roles of PCAF and Rad51 in DNA repair. Many results presented here are not entirely novel, including the effect of HDAC inhibition on HR and the identification of the acetylation site on Rad51. Several controls for essential findings are absent and validation for the experimental conditions used is not presented, making it difficult to assess the rigor and impact of the findings. Specific comments are listed below.

Response: We sincerely thank the reviewer for the time and valuable feedback on our manuscript. As suggested, we have performed the necessary experiments and revised the relative sections in the manuscript. We believe these revisions have significantly enhanced the clarity and overall quality of our work.

MAJOR:

1. PCAF regulation of HR is well known, importantly, this previous work has identified PCAF expression as a positive regulator of HR, not a suppressor of this repair pathway. For the results presented here to be of value to the research community some effort is needed to place these new findings within the context of what is known about PCAF and DNA repair, both experimentally and in the text as this was completely absent from this manuscript.

Response: Thank you very much for the insightful and constructive comments. As the reviewer noted, Kim et al. first demonstrated that PCAF depletion caused DSBs accumulation and HR deficiency in U2OS, HeLa, and HET293T cells. Interestingly, their data showed that the level of γ H2AX, a well-established DSB marker, was significantly decreased in PCAF KO U2OS cells and PCAF- Δ HAT overexpressed cells (Kim et al, 2019), which is assistant with our findings. Moreover, PCAF expression is consistently decreased in 5-fluorouracil (5-FU) resistant colorectal cancer (CRC), and knockdown of PCAF in HCT116 CRC parental cell line also increases resistance to 5-FU, which is consistent with our findings (Liu et al, 2019). We have added this clarification to the discussion section of the manuscript to better contextualize our study (Line 333-337).

2. A citation should be included for PMID: 31753913, the study which first described a HR repair defect in PCAF depleted cells. Additionally, the citation for PMID: 32966758 on line 76 is incorrect, this study identified a role for PCAF in replication fork degradation and is not appropriate for the transcriptional roles of PCAF, which were described years if not decades prior.

Response: We thank the reviewer for the expert insight. As suggested, we have cited this first work to describe the function of PCAF in HR repair (Line 333-334) and corrected the citation for replication fork degradation and transcriptional roles for PCAF in the revised manuscript (Line 76-78).

3. Are levels of H2BK120ac changed in the cell lines and conditions used here given the established role of PCAF in this process?

Response: Yes, the increased in H2BK120ac levels were observed in PCAF-overexpressed HeLa cells after ETO treatment, which is consistent with the established role of PCAF in this process.

Figure for referees not shown.

4. It is surprising that the authors do not include any work building on their previous studies regarding regulation of HR and Rad51 levels by GAPDH, demonstrating that GAPDH KD works synergistically with PCAF overexpression for Rad51 levels would strengthen both of those claims.

Response: We sincerely appreciate the reviewer's insightful suggestion that bridges our current findings with previous research. As our study primarily focused on the effect of the identified acetyltransferase PCAF on the acetylation and function of RAD51, we did not examine the potential synergistic interaction between GAPDH and PCAF in this work. However, this remains an intriguing avenue for further research. In accordance with your suggestion, we have incorporated a detailed discussion of this aspect into our manuscript (Line 337-340).

5. Does PCAF overexpression cancer types also present with decreased Rad51 levels? Is the pathway described here relevant in cancer models?

Response: Thank you for bringing this to our attention. We have ectopically expressed PCAF in human non-small cell lung cancer A549 cells and human breast carcinoma MDA-MB-231 cells and observed a reduction in the protein levels of RAD51. This data has been added in **Expanded view Figure 3 B-C**.

6. Inhibition of DNA repair and sensitivity to DSBs by TSA treatment is well known (PMID: 17722998).

Response: We fully understand the reviewer's concerns. Previous studies have revealed that TSA treatment impairs DNA damage repair (Roos & Krumm, 2016; Zhang et al, 2007), which we have cited in the revised version (Line 109). In our study, we emphasize the mechanism that TSA treatment inhibits HR by changing RAD51 acetylation levels. In addition, to further support the role of RAD51 acetylation in HR, we repeated the HR reporter assay and found that in cells expressing the RAD51 acetylation mutant, TSA treatment did not reduce HR efficiency as it did in wild type cells. This result has been added in the revised manuscript (Figure EV4B).

7. Is the interaction between PCAF and Rad51 enhanced by DNA damage? All of the interaction studies presented in figure 2 appear to be in non-damaged conditions, which may be less relevant to the pathway described.

Response: Thank you for the instructive suggestion. Given the critical role of RAD51 acetylation in response to DNA damage, we have investigated the interaction between endogenous RAD51 and exogenous PCAF and found increased interaction following ETO treatment. Furthermore,

immunofluorescence also showed the PCAF co-localized with RAD51 after ETO treatment, providing additional evidence that the interaction between PCAF and RAD51 is enhanced under DNA damage conditions. These data have been added in the revised manuscript (Figure 2E and EV2F-G).

8. Figures 3B and 6B do not include a blot for endogenous PCAF levels, the strength of these results is difficult to determine without knowing the expression level of FLAG-PCAF relative to wild type. These experiments should be done side by side, the level of PCAF overexpression needed to achieve Rad51 loss and HR suppression is critical to understand the impact of this claim.

Response: We agree with the reviewer that distinguishing endogenous and exogenous proteins is essential. However, due to the minimal molecular weight difference (~1 kDa) between exogenous and endogenous PCAF, we obtained only a single band when incubated with the PCAF antibody in Figure 3B. And we have added the immunoblot of PCAF for Figures 3B and 6B in the revised manuscript.

9. It is concerning that the Rad51 K40Q mutant partially reduces γ H2AX foci in Rad51 KO despite lower levels of Rad51 foci formation. It is also concerning that a Rad51 KO is used in this study, as it has been previously identified as an essential gene for cell proliferation (PMID: 33095861). No blot or qPCR is provided to validate this cell line.

Response: We understand the comment of the reviewer. In Figure 4E, although the RAD51-K40Q mutant formed fewer RAD51 foci, it was still able to function partially, thus it reduces γ H2AX foci in RAD51 KO cells.

Yes, the RAD51-deficient cells exhibited a reduced growth rate compared to WT cells. To generate the RAD51 KO cell line, we employed the CRISPR/Cas9 system followed by Blasticidin S selection. Initial Western blot failed to detect RAD51 at short exposure times. According to your suggestion, we have extracted the genomic DNA and performed sequencing. The results showed that the cell was heterogeneous. Furthermore, upon re-examining the cell lysates with prolonged exposure, we detected low levels of RAD51 protein, suggesting a partial knockdown rather than knockout (Figure EV4C). Accordingly, we have revised the relevant descriptions in the manuscript.

10. The role of RAD51 acetylation in regulating DNA repair is not very clear. Although phenotypes are correlated, how does acetylation of RAD51 promote these activities molecularly and mechanistically? As is, this is more observational than providing mechanistic understanding for these findings. For example, does acetylation of RAD51 play no role in HR repair? Is overexpression of PCAF then just a detrimental consequence of these conditions? It seems more likely that this acetylation is regulatory and plays roles in normal repair but when dysregulated, because detrimental to repair. A clearer message from the data provided needs to be synthesized into a molecular and mechanistic model to understand the relevance and impact of these findings.

Response: Thank you for the comment. As you mentioned, the acetylation of RAD51 plays a regulatory role in HR repair, and its timely removal from chromatin is crucial for the completion of HR. Previous studies have already showed that RFW3-mediated ubiquitination contributes to the timely removal of RAD51 from DNA damage sites to facilitate HR (Inano et al, 2017).

However, the mechanism by which this E3 ligase recognizes RAD51 remains unclear. Our work established a link between RAD51 acetylation and ubiquitin-mediated degradation, which we have discussed in the discussion section (Line 294-301).

MINOR

1. An un-damaged control would be useful in figure 1B, to demonstrate that Rad51 chromatin association is increased after damage.

Response: We have now moved the original Figure EV1B to the main figure panel as Figure 1B and revised the corresponding text in the results section (Line 98-100).

2. In the blots presented in figure 3 E and F the changes in Rad51 levels look visually less than the accompanying quantification. Lower exposures for these blots might better highlight the differences in protein levels.

Response: Thank you for your suggestion. We have repeated the experiment and obtained qualified images. These updated results have been incorporated into the revised manuscript (Figure 3E and 3F).

Referee #3:

In this manuscript, Hou and colleagues performed several experiments to examine the mechanism and the function of RAD51 acetylation during DSB repair. They show that RAD51 interacts with and is acetylated by PCAF. They map the interaction domains between both proteins, identify a key residue that is likely acetylated by PCAF and suggest that RAD51 acetylation by PCAF promotes its proteasome-mediated degradation. Lastly, they demonstrate a requirement for the acetylation domain of PCAF and suggest that acetylation of RAD51 by PCAF downregulates homologous recombination (HR) by increasing RAD51 turnover.

Overall, the work is interesting and increases our understanding of the modifications regulating the stability, localization and activity of RAD51. The data is also largely well-presented, and the questions have been extensively investigated. However, the main problem with the work is that some key data show relatively minor differences that, while significant in some cases, may not be critical for RAD51 and HR regulation especially since a lot of the phenotypes stem from PCAF overexpression. This weakness feeds into another problem in which there are quite a few instances where the data have been overinterpreted. To strengthen the manuscript, an overarching mechanism for how this potentially key role of PCAF in controlling RAD51 levels is regulated by DDR and how it integrates with known mechanisms of RAD51 degradation like RFD3 is missing or inadequate and ought to be addressed.

Response: Thank you very much for reviewing our manuscript. We really appreciate your time and thoughtful comments. In response to your suggestions, we have performed the necessary experiments to address your concerns and now we believe all the concerns are resolved.

Concerns

The changes in Fig 1B are not readily evident from the blot and needs to be quantified.

Response: Thank you for your suggestion. We have added the quantification in the revised manuscript (Figure 1C).

While TSA treatment reduces HR in Figure 1E, that is not necessarily related to Rad51. The authors need to repeat this experiment in a later figure showing that the RAD51 acetylation mutant doesn't show diminished HR upon TSA treatment.

Response: Thank you for the expert insights. To further investigate the role of RAD51 acetylation in HR, we examined the HR efficiency in U2OS cells transfected with different RAD51 mutants, with or without TSA treatment (Figure EV4B). Our results showed that TSA treatment downregulated HR efficiency by approximately 40% in both the control and siRAD51+RAD51-WT groups. However, in the siRAD51, siRAD51+RAD51-K40R, and siRAD51+RAD51-K40Q groups, HR efficiency was only reduced by 10% under TSA treatment. These findings suggest that although RAD51 acetylation is crucial for HR, it may not be the sole factor contributing to the downregulation of HR efficiency after TSA treatment. The roles of other HR-related proteins, such as ATM, BARD1, which are also known to be regulated by acetylation (Blackford & Jackson, 2017; Minten et al, 2021) .

How is this pathway regulated and controlled by DDR? In Fig 2E, we already see significant interaction in the absence of drug and is definitely not 'greatly enhanced' (line 131). The 'increase' appears to be due to reduced RAD51 levels in the input, PCAF levels appear unchanged. Also, 2E is better interpreted with endogenous PCAF IP'ed with RAD51.

Response: We appreciate this critical analysis. Previous studies have demonstrated that PCAF is phosphorylated by ATR at S264 (Kim et al, 2020). Given that ATR functions as an upstream kinase in the HR pathway, whether this phosphorylation modulates the function of PCAF in the HR process remains to be further confirmed.

The quantification presented in Figure 2E represents the ratio of Flag-PCAF to the pulled-down RAD51 in the IP group. Given that RAD51 protein levels varied in the input group, this ratio more accurately reflects changes in the interaction between PCAF and RAD51 following ETO treatment. However, the use of 'greatly increased' to describe this result may have been an overstatement. Accordingly, we have revised it to 'increased' in the updated manuscript (Line 138-139).

As you pointed out, the results from endogenous PCAF immunoprecipitation would offer stronger support for our conclusion. However, due to the unsuitability of the anti-PCAF antibody for IP, we have therefore chosen to use endogenous RAD51 in our experiments.

Does PCAF colocalize with RAD51? A quick IF can address this.

Response: We fully agree with your suggestion and have accordingly performed an immunofluorescence assay, which revealed significant co-localization of PCAF and RAD51 following ETO treatment (Figure EV2G).

Figure 4G shows very weak differences that is not consistent with the interpretation. The only clearcut differences are with RAD51 KO.

Response: We understand the reviewer's concern and have repeated the cell counting kit-8 assay to address this issue. We found that the previous subtle differences were due to the drug dosage.

After increasing the drug concentration, the differences between groups become more pronounced. The updated data have been included in the revised manuscript (Figure 4G).

Indices of significance are lacking in Figure 5A

Response: Thank you for pointing this out. In Figure 5A, the column represents the median values derived from the GEPIA database (<http://gepia.cancer-pku.cn/>). Since GEPIA does not provide statistical significance values, we reanalyzed the expression of PCAF in tumors using the UALCAN (the University of ALabama at Birmingham CANcer data analysis portal) database (<https://ualcan.path.uab.edu/analysis.html>), and the results were consistent with those obtained from GEPIA. Therefore, we have replaced this result with the analysis data from UALCAN (Figure 5A).

Figure 5 will be strengthened by performing the PCAF KD using the HR reporter.

Response: Thank the reviewer for the valuable suggestion. We have examined the HR efficiency and observed higher efficiency in PCAF knockdown cells. The new data have been added in the updated manuscript (Figure 5G).

In Fig 5, at later time points, 8 hrs and 24 hrs post etoposide treatment, there is a significantly higher number of RAD51 foci in PCAF overexpressing cells compared to EV control. Similarly, PCAF knockdown cells have fewer RAD51 foci than sgNC cells 4 hrs and 8 hrs post Etoposide treatment. How do the authors reconcile this?

Response: We understand the reviewer's concerns about the higher number of RAD51 foci at later time points. PCAF overexpression reduced RAD51 protein levels, resulting in decreased RAD51 foci formation and impaired HR at early stages of ETO treatment. However, at later time points, the control group exhibited fewer RAD51 foci, indicating that HR was nearly complete. In contrast, PCAF overexpressing cells retained more γ H2AX foci due to impaired HR, which required continued recruitment of RAD51 to complete DNA repair. Consequently, these cells displayed more RAD51 foci compared to the control group. A same trend was also observed in the sgNC and sgPCAF groups. We have also added a detailed explanation of this phenomenon in the revised manuscript (Line 220-222).

What is going on in Fig 6C? The figure is not consistent with the quantification. Based on the blot shown, the level of ubiquitination in the cells transfected with EV is similar to the PCAF-WT OE. More importantly, RAD51's size is unchanged in the ubiquitinated blot (the most prominent band is at 37KD even though ubiquitin is 8.5KD). This experiment (and the ones in Figure 3G and H) are not convincing as indicative of RAD51 ubiquitination and is better done in well controlled in vitro conditions.

Response: We apologize for the incorrect labeling of the Y-axis in Figure 6C, the "Relative protein level of Ubi" should actually be "Ratio of Ubi/RAD51". Although the ubiquitination level in the WT group was slightly higher than that in the EV group, the RAD51 level in the WT group was lower, leading to a higher Ubi/RAD51 ration in the WT group compared to the EV group. We agree with the reviewer that detecting RAD51 ubiquitination under in vitro conditions would be better. However, considering that the specific E3 ubiquitin ligase involved in the process is still unclear, it's difficult to perform the in vitro ubiquitination assay. To further clarify the RAD51

ubiquitination, we have co-transfected His-Ubi with Flag-PCAF, sgPCAF or Flag-PCAF-ΔHAT into HEK293T cells. By performing a His-Ubi pull-down followed by Western blotting with the anti-RAD51 antibody, we obtained high-quality ubiquitinated RAD51 bands. The updated images have been included in the revised version (Figure 3G, 3J and Appendix Figure S1A).

In Fig 6H, the authors should examine the cell cycle as a control.

Response: Thank you. The previous study showed that PCAF impairs cell proliferation by causing S and G2/M cell cycle arrest (Mateo et al, 2009). Since HR occurs exclusively in the S and G2 phases, we propose that the reduction in HR efficiency caused by PCAF overexpression is not due to cell cycle arrest.

Based on your suggestion, we analyzed the cell cycle in control cells, as well as in cells overexpressing Flag-PCAF-WT or Flag-PCAF-ΔHAT over-expressed cells. Our results showed that ectopic expression of PCAF led to increase in the S and G2/M phases, whereas PCAF-ΔHAT did not (Appendix Figure S1G). These findings confirm that the impairment of HR by PCAF is independent of its effect on the cell cycle.

Is knockdown of RFWD3 additive or synergistic with sgRNA to PCAF?

Response: We appreciate the reviewer's insightful suggestion. It has been reported that RFWD3 mediated ubiquitination plays a key role in the timely removal of RAD51 from DNA damage sites (Inano et al, 2017). Consistent with their finding, our data show that acetylated RAD51 is more susceptible to ubiquitination and degradation, providing a possible explanation for the specific recognition of RAD51 by RFWD3. Therefore, knockdown of RFWD3 is likely to synergize with PCAF, and this requires further investigation in future studies.

Minor

Title of Supp 1 is misleading (they're not looking at acetylation)

Response: Thank you for pointing out this issue. We have revised the title of Supplementary Figure 1 to better reflect its content (Line 864).

Line 100 is a stretch. There's no link as of yet between acetylation and stability. The authors can say something like we wanted to investigate the possibility.

Response: Thank you. We have reorganized the language according to your suggestions (Line 97-98).

The figures referenced in discussion should be cited.

Response: We thank the reviewer for pointing this out and have cited the relevant figures in the discussion section.

References:

Blackford AN, Jackson SP (2017) ATM, ATR, and DNA-PK: The trinity at the heart of the DNA damage response. *MOL CELL* **66**(6): 801-817

Inano S, Sato K, Katsuki Y, Kobayashi W, Tanaka H, Nakajima K, Nakada S, Miyoshi H, Knies K, Takaori-Kondo A, Schindler D, Ishiai M, Kurumizaka H, Takata M (2017) RFD3-Mediated ubiquitination promotes timely removal of both RPA and RAD51 from DNA damage sites to facilitate homologous recombination. *MOL CELL* **66**(5): 622-634

Kim JJ, Lee SY, Choi JH, Woo HG, Xhemalce B, Miller KM (2020) PCAF-Mediated histone acetylation promotes replication fork degradation by MRE11 and EXO1 in BRCA-Deficient Cells. *MOL CELL* **80**(2): 327-344

Kim JJ, Lee SY, Gong F, Battenhouse AM, Boutz DR, Bashyal A, Refvik ST, Chiang CM, Xhemalce B, Paull TT, Brodbelt JS, Marcotte EM, Miller KM (2019) Systematic bromodomain protein screens identify homologous recombination and R-loop suppression pathways involved in genome integrity. *GENE DEV* **33**(23-24): 1751-1774

Liu T, Wang X, Hu W, Fang Z, Jin Y, Fang X, Miao QR (2019) Epigenetically Down-regulated acetyltransferase PCAF increases the resistance of colorectal cancer to 5-fluorouracil. *NEOPLASIA* **21**(6): 557-570

Mateo F, Vidal-Laliena M, Canela N, Zecchin A, Martinez-Balbas M, Agell N, Giacca M, Pujol MJ, Bachs O (2009) The transcriptional co-activator PCAF regulates cdk2 activity. *NUCLEIC ACIDS RES* **37**(21): 7072-7084

Minten EV, Kapoor-Vazirani P, Li C, Zhang H, Balakrishnan K, Yu DS (2021) SIRT2 promotes BRCA1-BARD1 heterodimerization through deacetylation. *CELL REP* **34**(13): 108921

Roos WP, Krumm A (2016) The multifaceted influence of histone deacetylases on DNA damage signalling and DNA repair. *NUCLEIC ACIDS RES* **44**(21): 10017-10030

Zhang Y, Carr T, Dimtchev A, Zaer N, Dritschilo A, Jung M (2007) Attenuated DNA damage repair by trichostatin A through BRCA1 suppression. *RADIAT RES* **168**(1): 115-124

Dear Dr. Pan,

Thank you for the submission of your revised manuscript. We have now received the enclosed reports from the referees as well as referee cross-comments, all pasted below.

As you will see, both referees 2 and 3 still have remaining concerns that I would like you to address and incorporate (along the lines suggested by all referees) before we can proceed with the official acceptance of your manuscript.

A few editorial requests will also need to be addressed:

- Please correct the REFERENCE style to the EMBO reports style: et al needs to be used after 10 author names.
- Please submit with your final ms a completed author checklist, which you can download from our author guidelines <<https://www.embopress.org/page/journal/14693178/authorguide>>. The completed author checklist will also be part of the transparent peer-review file.
- The FUNDING INFO is missing in the ms file - it needs to be part of the Acknowledgments.
- A FIGURE CALLOUT is missing for Figure 2B in the ms text, please add.
- The Methods section needs to include a separate Reagents and Tools Table file (listing key reagents, experimental models, software and relevant equipment and including their sources and relevant identifiers) followed by a Methods and Protocols section in which methods are described using a step-by-step protocol format with bullet points. A downloadable templates (.docx) for the Reagents and Tools Table can be found in our author guidelines: <<https://www.embopress.org/page/journal/14693178/authorguide#manuscriptpreparation>>.
- Materials and Methods should be just Methods
- The nomenclature of the EV figure legends should be Figure EV1, etc. instead of Expanded view Figure 1, etc. Please correct.
- In our routine image analysis of to-be-accepted ms, we detected several boxes with no signal:

Figure 4E. SD (Source Data) provided 0 pixel value

Figure 6E. SD Provided 2 cells 0 Pixel value

Figure EV1B.

Figure EV2F and G.

Figure EV4E.

Figure EV5 B,D,E&F

Appendix Fig. S1F

Can you please provide source data for all these panels and asses the figures. If the panel/image contains no signal - 0 pixel count - then a cross will need to be added to the panel showing that no signal was able to be captured. Usually some background signal is present in microscopy images.

- Please provide exact p values (as reasonable) in the legends of figures 1A, C, D, E, F, G, H; 2E, 3A, B, C, D, E, F, I, L; 4A, B, C, F, G; 5A, C, D, E, F, G, H, I, J, K, L; 6A, C, D, E, F, G, H, I; EV3 A-F; EV4 B, F, G, H; EV5 G-I S1B, C, E, F, G, H.

EMBO press papers are accompanied online by A) a short (1-2 sentences) summary of the findings and their significance, B) 2-3 bullet points highlighting key results and C) a synopsis image that is exactly 550 pixels wide and 200-600 pixels high (the height is variable). The synopsis image should provide a sketch of the major findings, like a graphical abstract. Please note that text needs to be readable at the final size. Please send us this information along with the final manuscript.

Referee #1:

The authors have tried their best addressing my comments.

The study should be of interest to the community.

Referee #2:

This revision unfortunately does not address previous concerns and only raises new ones. While it is appreciated that the authors provided new data and revisions in response to comments, this work is not rigorous or complete enough to warrant publication in EMBO Reports. Major issues remain. This includes the immunofluorescence data. Almost all of the RAD51 IF data seems to be non-specific or non-damage (doesn't colocalize with known markers of DNA damage like phosphoH2AX). For example, new data in supplemental figure 2 F and G purports to show PCAF localization with RAD51. These data appear to show no colocalization and there is no quantification to these data and no controls. Similar issues are found throughout this manuscript. Another issue is the inconsistency with published data. The following was previously asked:

1. PCAF regulation of HR is well known, importantly, this previous work has identified PCAF expression as a positive regulator of HR, not a suppressor of this repair pathway. For the results presented here to be of value to the research community some effort is needed to place these new findings within the context of what is known about PCAF and DNA repair, both experimentally and in the text as this was completely absent from this manuscript.

Response: Thank you very much for the insightful and constructive comments. As the reviewer noted, Kim et al. first demonstrated that PCAF depletion caused DSBs accumulation and HR deficiency in U2OS, HeLa, and HET293T cells. Interestingly, their data showed that the level of H2AX, a well-established DSB marker, was significantly decreased in PCAF KO U2OS cells and PCAF- HAT overexpressed cells (Kim et al, 2019), which is assistant with our findings. Moreover, PCAF expression is consistently decreased in 5-fluorouracil (5-FU) resistant colorectal cancer (CRC), and knockdown of PCAF in HCT116 CRC parental cell line also increases resistance to 5-FU, which is consistent with our findings (Liu et al, 2019). We have added this clarification to the discussion section of the manuscript to better contextualize our study (Line 333-337). This response is not satisfactory. 5-FU is an alkylating agent and this question is about DSB repair. PCAF deficiency has been shown to reduce HR repair (Kim et al. 2019). Here, the authors claim the opposite and say that loss of PCAF increase HR repair. No effort is given to address these discrepancies.

There are other instances where the data does not support the model (which was also pointed out by other reviewers). For example, in TSA treated samples, it is shown that RAD51 degrades more than untreated cells after CHX treatment. If RAD51 acetylation promotes degradation, then increasing acetylation by TSA treatment should result in increase degradation that is rescued by CHX treatment. This doesn't seem to be the result. In other CHX experiments, the results are similarly difficult to interpret and not entirely consistent with the proposed model.

Taken together, this work has too many issues, several of which are provided above, for publication.

Referee #3:

The authors have largely addressed my concerns in their revised manuscript. I do however ask that the authors reconcile a concern about the conflicting data on the role of PCAF in homologous recombination initially mentioned by Reviewer 2. Previous data (PMID: 32966758) showed opposite effects to their current findings and this should be addressed directly with some plausible explanations. Their current discussion of this puzzling contradiction is insufficient and is almost glossed over. Once this is addressed, I recommend publication of the manuscript in EMBO reports.

Cross-comments from referee 1:

Ask authors to rigorously quantify the cell biological data, include more appropriate controls, to support conclusions of protein co-localization. This is a valid point of Reviewer #2.

Also address (textual changes OK) why opposite conclusions re. PCAF's role in DSB repair are drawn compared to Kim et al (Mol Cell, PMID: 32966758). This is Reviewer #3's rec too.

Cross-comments from referee 3:

I have read reviewer 2's concerns and as you can see, the concern on properly reconciling or addressing the variance in their data with prior published work mirrors mine. This can be addressed in the text.

Experimentally, I also would like to know which RAD51 antibody they're using to examine endogenous RAD51 because the foci don't look like you'd expect (and these are supposedly the 'best' images). Also they should add quantification to the new Supp figures 2F and 2G. I should note that a lot of their imaging conclusions are made using Flag-tagged WT and mutant Rad51 and those look good to me. I don't share the concern with the CHX experiment - I'm not sure that CHX would rescue the increased degradation upon TSA treatment.

Overall, there are some minor issues with rigor and model fit but I think since reviewer 1 is also in agreement, the paper can be accepted with minor revisions.

Referee #1:

The authors have tried their best addressing my comments.

The study should be of interest to the community.

Response: We would like to express our gratitude for your thorough review and valuable feedback. Thank you once again for your time, expertise, and positive evaluation of our work.

Referee #2:

This revision unfortunately does not address previous concerns and only raises new ones. While it is appreciated that the authors provided new data and revisions in response to comments, this work is not rigorous or complete enough to warrant publication in EMBO Reports. Major issues remain. This includes the immunofluorescence data. Almost all of the RAD51 IF data seems to be non-specific or non-damage (doesn't colocalize with known markers of DNA damage like phosphoH2AX). For example, new data in supplemental figure 2 F and G purports to show PCAF localization with RAD51. These data appear to show no colocalization and there is no quantification to these data and no controls. Similar issues are found throughout this manuscript.

Response: We appreciate the time and effort the reviewer invested in reviewing our manuscript. Our immunofluorescence data revealed partial co-localization of RAD51 foci with γ H2AX foci. In the enlarged fluorescence images shown in Figure EV2F and EV2G, yellow fluorescence, indicative of the co-localization of red fluorescence (PCAF or RAD51) and green fluorescence (γ H2AX), was observed. As you pointed out, not all RAD51 foci in our images co-localize with γ H2AX, which is consistent with other studies (Pinedo-Carpio et al, 2023; Tischler et al, 2024). This phenomenon may be related to the dynamic balance between the extent of DNA double-strand breaks and the expression level of RAD51. To more clearly demonstrate the co-localization, we have included the corresponding quantitative analysis (Fig EV2F-G).

Another issue is the inconsistency with published data. The following was previously asked:

1. PCAF regulation of HR is well known, importantly, this previous work has identified PCAF expression as a positive regulator of HR, not a suppressor of this repair pathway. For the results presented here to be of value to the research community some effort is needed to place these new findings within the context of what is known about PCAF and DNA repair, both experimentally and in the text as this was completely absent from this manuscript.

Response: Thank you very much for the insightful and constructive comments. As the reviewer noted, Kim et al. first demonstrated that PCAF depletion caused DSBs accumulation and HR deficiency in U2OS, HeLa, and HET293T cells. Interestingly, their data showed that the level of γ H2AX, a well-established DSB marker, was significantly decreased in PCAF KO U2OS cells and PCAF- Δ HAT overexpressed cells (Kim et al, 2019), which is assistant with our findings. Moreover, PCAF expression is consistently decreased in 5-fluorouracil (5-FU) resistant colorectal cancer (CRC), and knockdown of PCAF in HCT116 CRC parental cell line also increases resistance to 5-FU, which is

consistent with our findings (Liu et al, 2019). We have added this clarification to the discussion section of the manuscript to better contextualize our study (Line 333-337).

This response is not satisfactory. 5-FU is an alkylating agent and this question is about DSB repair. PCAF deficiency has been shown to reduce HR repair (Kim et al. 2019). Here, the authors claim the opposite and say that loss of PCAF increase HR repair. No effort is given to address these discrepancies.

Response: Thanks for your expert insights. We acknowledge that there are some differences between our conclusions and those of Kim et al. Kim et al. demonstrated that PCAF participates in the HR process by acetylating histones, showing that PCAF knockout impairs HR and thereby identifying PCAF as a positive regulator of HR. In contrast, our study found that PCAF knockdown reduces RAD51 acetylation, leading to increased protein stability and enhanced HR, while PCAF overexpression enhances RAD51 acetylation and promotes its premature dissociation from chromatin, further highlighting the dynamic regulatory role of PCAF in HR proteins. Although there are some inconsistencies between our findings and those of Kim et al., they are not contradictory. It is well established that the dissociation of HR proteins from chromatin is essential for the completion of HR (Inano et al, 2017). In the absence of PCAF, RAD51 may fail to undergo acetylation and consequently be unable to dissociate from chromatin, thereby impeding HR progression. From this perspective, the findings of Kim et al. indeed support the critical role of PCAF in HR. Importantly, our experiments involved PCAF knockdown rather than complete knockout, resulting in residual PCAF expression within the cells. This residual PCAF may be sufficient to facilitate RAD51 acetylation and its timely dissociation from chromatin. Therefore, it is plausible that an optimal level of PCAF is crucial for the proper completion of HR, as both excessive and insufficient PCAF could alter the chromatin binding dynamics of RAD51 and compromise HR efficiency. This precise regulation of protein levels is an important aspect that warrants further investigation. We have incorporated a relevant discussion in the revised manuscript (Line 330-346).

There are other instances where the data does not support the model (which was also pointed out by other reviewers). For example, in TSA treated samples, it is shown that RAD51 degrades more than untreated cells after CHX treatment. If RAD51 acetylation promotes degradation, then increasing acetylation by TSA treatment should result in increase degradation that is rescued by CHX treatment. This doesn't seem to be the result. In other CHX experiments, the results are similarly difficult to interpret and not entirely consistent with the proposed model.

Taken together, this work has too many issues, several of which are provided above, for publication.

Response: Thank you for the comment. In this experiment, CHX treatment was not intended to rescue protein degradation but rather to inhibit protein synthesis, allowing us to compare RAD51 protein stability before and after acetylation modification. Therefore, the accelerated degradation of RAD51 observed in the TSA-treated group supports the conclusion that acetylation enhances RAD51 degradation.

Referee #3:

The authors have largely addressed my concerns in their revised manuscript. I do however ask that the authors reconcile a concern about the conflicting data on the role of PCAF in homologous recombination initially mentioned by Reviewer 2. Previous data (PMID: 32966758) showed opposite effects to their current findings and this should be addressed directly with some plausible explanations. Their current discussion of this puzzling contradiction is insufficient and is almost glossed over. Once this is addressed, I recommend publication of the manuscript in EMBO reports.

Response: We truly appreciate your contributions for improving the qualities of our manuscript. Based on your suggestions, we carefully revisited the study by Kim et al. and found that, although there are apparent inconsistencies between our findings, they are not necessarily contradictory. We propose that the acetylation of RAD51 by PCAF is essential for its timely dissociation from chromatin. Notably, the findings of Kim et al. can also be interpreted in support of our model. In their experiments, they employed either complete knockout or highly efficient knockdown of PCAF. Consequently, when the intracellular level of PCAF is markedly reduced, RAD51 fails to undergo acetylation and subsequently dissociate from chromatin in a timely manner, thereby impairing HR progression. In contrast, the knockdown efficiency of PCAF in our experiments was relatively low, resulting in the presence of a moderate amount of residual PCAF within the cells. This remaining PCAF may be sufficient to acetylate RAD51 and facilitate its timely dissociation from chromatin. Therefore, maintaining an optimal level of PCAF appears to be crucial for efficient HR, as both excessive and insufficient PCAF could disrupt the chromatin-binding dynamics of RAD51 and hinder HR progression. This fine-tuned regulation of protein levels represents an important aspect that warrants further investigation. We have incorporated the relevant discussion into the revised manuscript (Line 330-346).

Cross-comments from referee 1:

Ask authors to rigorously quantify the cell biological data, include more appropriate controls, to support conclusions of protein co-localization. This is a valid point of Reviewer #2.

Response: We appreciate your insightful suggestion and we have added the corresponding quantitative analysis in our manuscript (Fig EV2F-G).

Also address (textual changes OK) why opposite conclusions re. PCAF's role in DSB repair are drawn compared to Kim et al (Mol Cell, PMID: 32966758). This is Reviewer #3's rec too.

Response: Thank you for your suggestion. We have provided a plausible explanation for the discrepancies between our findings and those of Kim et al. in the revised manuscript (Line 330-346).

Cross-comments from referee 3:

I have read reviewer 2's concerns and as you can see, the concern on properly reconciling or addressing the variance in their data with prior published work mirrors mine.

This can be addressed in the text.

Response: Thanks for your kind comments. We have added a discussion of the differences in the result between our study and prior published work in the revised manuscript (Line 330-346).

Experimentally, I also would like to know which RAD51 antibody they're using to examine endogenous RAD51 because the foci don't look like you'd expect (and these are supposedly the 'best' images). Also they should add quantification to the new Supp figures 2F and 2G. I should note that a lot of their imaging conclusions are made using Flag-tagged WT and mutant Rad51 and those look good to me. I don't share the concern with the CHX experiment - I'm not sure that CHX would rescue the increased degradation upon TSA treatment.

Response: We sincerely appreciate your comments. The antibody used to detect RAD51 foci was purchased from Abcam (Cat#ab133534).

Overall, there are some minor issues with rigor and model fit but I think since reviewer 1 is also in agreement, the paper can be accepted with minor revisions.

Response: We sincerely appreciate the reviewer's thorough evaluation and constructive feedback on our manuscript. We have completely revised our article based on these comments.

References:

Inano S, Sato K, Katsuki Y, Kobayashi W, Tanaka H, Nakajima K, Nakada S, Miyoshi H, Knies K, Takaori-Kondo A et al. (2017) RFWD3-Mediated Ubiquitination Promotes Timely Removal of Both RPA and RAD51 from DNA Damage Sites to Facilitate Homologous Recombination. *MOL CELL* **66**(5): 622-634

Pinedo-Carpio E, Dessapt J, Beneyton A, Sacre L, Berube MA, Villot R, Lavoie EG, Coulombe Y, Blondeau A, Boulais J et al. (2023) FIRRM cooperates with FIGNL1 to promote RAD51 disassembly during DNA repair. *SCI ADV* **9**(32): eadf4082

Tischler JD, Tsuchida H, Bosire R, Oda TT, Park A, Adeyemi RO (2024) FLIP(C1orf112)-FIGNL1 complex regulates RAD51 chromatin association to promote viability after replication stress. *NAT COMMUN* **15**(1): 866

Dear Dr. Pan,

Thank you for the submission of your revised manuscript and for your explanation for the figure panels with empty cells.

We understand that some figure panels show no signal, and your explanation is fine. Please include this explanation in your ms file somewhere (if it is not in there yet) and please also label all main figure panels with no signal with an X placed across the entire figure panel and please state in the figure legend that with the chosen microscopy settings, no signal was obtained. This needs to be done for all relevant figures (Figure 4E, Figure 6E, Figure EV1B, Figure EV2F and G, Figure EV4E, Figure EV5 B,D,E&F, Appendix Fig. S1F).

I am making another decision on your ms now so that you can upload new files. You should be able to bring forward all files from this version of the ms to the new version and you can then replace only the files that need replacement.

The Source Data (SD) for these figures need to be part of their corresponding SD folder, no extra SD folder is needed. No X is needed in the SD images.

The synopsis image you sent is OK but the text on the image is too small. Can you please send us a new synopsis image with larger text? Thank you.

I also would like to suggest to modify the ms title to make it a little more specific. What do you think about this:

PCAF-mediated acetylation removes RAD51 from chromatin to facilitate late-phase HR repair

And the last sentence of the abstract could be :

Our results highlight a novel role of PCAF in HR and provide a possible mechanism for tumor development and drug resistance caused by low expression of PCAF.

I look forward to seeing a new, final version of your manuscript as soon as possible.

Referee #3:

The authors have addressed my concerns and I recommend publication.

9-Jun-2025

Revised manuscript EMBOR-2024-60714V3

Esther Schnapp, PhD

Editor

EMBO Reports

RE: Manuscript EMBOR-2024-60714V3 " PCAF-Mediated Acetylation Regulates RAD51 Dynamic Localization on Chromatin during HR Repair "

Dear Dr. Schnapp,

Thank you for your e-mail message regarding our revised manuscript EMBOR-2024-60714V3. We are very pleased that the reviewers have agreed to the publication of our manuscript.

In accordance with your suggestion, we have marked all main figure panels showing no signal with an "X" across the entire panel, and this has been clearly stated in the corresponding figure legends. A detailed explanation has also been added to the Methods section. Additionally, we have adjusted the font size in the synopsis image to enhance readability and improve overall clarity. All relevant revised files and folders have been updated accordingly.

Thank you very much for your thoughtful suggestion regarding the modification of the manuscript title. While the revised title is indeed more specific and highlights the positive regulatory role of PCAF in HR, after careful consideration, we believe that the original title more accurately reflects the central focus of our study, particularly the dynamic regulatory role of PCAF in HR proteins. As discussed in the manuscript, an optimal level of PCAF is crucial for the proper completion of HR, as both excessive and insufficient PCAF can alter the chromatin binding dynamics of RAD51 and compromise HR efficiency. Therefore, we respectfully prefer to retain the original title and hope our rationale is clear and acceptable.

We hope that the revised version is now deemed suitable for publication in *EMBO Reports* and look forward to your final decision.

Sincerely,

Feiyan Pan, Ph.D.

College of Life Sciences,

Nanjing Normal University

1 Wenyuan Rd., Nanjing, China 210023

E-mail: panfeiyan@njnu.edu.cn

Dr. Feiyan Pan
Nanjing Normal University
College of Life Sciences
Nanjing, Jiangsu
China

Dear Dr. Pan,

I am very pleased to accept your manuscript for publication in the next available issue of EMBO reports. Thank you for your contribution to our journal.

Yours sincerely,
